# Roxadustat as a Hypoxia-Mimetic Agent: Erythropoietic Mechanisms, Bioanalytical Detection, and Regulatory Considerations in Sports Medicine

**DOI:** 10.3390/cimb47090734

**Published:** 2025-09-09

**Authors:** Elena-Christen Creangă, Cristina Ott, Alina-Crenguţa Nicolae, Cristina Manuela Drăgoi, Raluca Stan

**Affiliations:** 1Faculty of Chemical Engineering and Biotechnology, National University of Science and Technology Politehnica Bucharest, Str. Gheorghe Polizu, nr. 1-7, Sector 1, 011061 Bucharest, Romania; elena.creanga@upb.ro (E.-C.C.); cristina.ott@upb.ro (C.O.); raluca.stan@upb.ro (R.S.); 2Faculty of Pharmacy, “Carol Davila” University of Medicine and Pharmacy, 6 Traian Vuia St., 020956 Bucharest, Romania; cristina.dragoi@umfcd.ro

**Keywords:** roxadustat, erythropoietin, chronic kidney disease, hypoxia-inducible factor prolyl hydroxylase inhibitor, bioanalysis, Quechers method

## Abstract

Roxadustat (ROX) is an orally active inhibitor of hypoxia-inducible factor prolyl hydroxylase (HIF-PHI) that exerts erythropoietic, cardioprotective, and metabolic regulatory effects. Approved for the treatment of anemia associated with chronic kidney disease, ROX promotes endogenous erythropoietin production and improves iron homeostasis, providing a non-injectable alternative to conventional erythropoiesis-stimulating agents (ESAs). Its ability to enhance oxygen transport and facilitate muscle recovery has, however, led to its misuse in sports, where it is classified as a banned substance by the World Anti-Doping Agency. This review provides a comprehensive overview of the pharmacological properties of ROX, its approved and investigational clinical applications, and its chemical synthesis strategies. Particular emphasis is placed on the analytical methodologies employed for ROX detection in anti-doping settings. Techniques such as liquid chromatography–tandem mass spectrometry (LC–MS/MS), ultraviolet–visible (UV–Vis) spectroscopy, Fourier-transform infrared spectroscopy (FT-IR), and high-performance thin-layer chromatography (HPTLC) are critically assessed for their efficacy in detecting ROX and its metabolites in biological matrices. Given the increasing incidence of ROX misuse among athletes, ongoing optimization of detection protocols and longitudinal monitoring approaches, are essential to uphold both sports integrity and public health.

## 1. Overview

Hypoxia-inducible factor prolyl-hydroxylase inhibitors (HIF-PHIs) are small-molecule inhibitors of the HIF prolyl-hydroxylase domain (PHD) enzymes that normally hydroxylate specific proline residues on HIF-α under normoxia, marking HIF-α for pVHL-mediated ubiquitination and proteasomal degradation. By transiently inhibiting PHD activity, HIF-PHIs stabilize HIF-1α and HIF-2α, enabling nuclear accumulation, dimerization with HIF-β (ARNT), and binding to hypoxia-response elements (HREs) to drive the transcription of genes that regulate erythropoiesis and iron metabolism, among other adaptive programs [1,2]. This framework is grounded in foundational discoveries on HIF-1 as an O_2_-regulated bHLH-PAS heterodimer (Semenza and colleagues), which established the central role of HIF signaling in cellular oxygen sensing and gene regulation [3].

Quantitative hypoxia/HIF-PHI responses: HIF stabilization drives robust EPO induction with defined magnitudes and kinetics. In vivo, roxadustat increased serum EPO within hours in rodent models, consistent with pharmacodynamic expectations [4]. In rats, roxadustat markedly increased Epo mRNA in both kidney and liver, while Epo protein accumulated predominantly in the kidney; Epo mRNA rose by ~700–1100× in the kidney and ~200–330× in the liver versus baseline, depending on dose [5]. Complementarily, short-course roxadustat pretreatment increased renal Epo before ischemic injury (~6–9×), and hypoxia alone induced ~75× increases, illustrating the dynamic range of the response [6]. In human Hep3B cells, roxadustat induces EPO mRNA and secreted EPO in a concentration-dependent manner, confirming direct pharmacologic activation at the EPO locus [4].

Single-cell/lineage evidence for renal EPO-producing cells: Recent lineage-tracing and single-cell studies localize EPO production to rare PDGFRβ+ interstitial fibroblasts (renal EPO-producing, REP cells). Lineage labeling demonstrates that only a small fibroblast subset in the renal interstitium produces Epo in vivo [7]; foundational work isolated REP cells as fibroblast-like interstitial cells comprising ~0.2% of whole kidney cells [8]. REP-like cell models generated by conditional tagging confirm HIF-2α-dominant Epo induction ex vivo [9]. Furthermore, EPO induction in the kidney—not the liver—appears essential for the therapeutic effect of HIF activators in renal anemia [10]. These data complement the mechanistic framework already cited (PHD inhibition → HIF-α stabilization → HIF-α/β dimerization → HRE binding) and support a primary role for HIF-2α in erythropoietic gene control [11].

Erythropoiesis represents a highly regulated and intricate physiological process responsible for the production of red blood cells. This process is critically dependent on the availability of erythropoietin (EPO), a glycoprotein hormone primarily synthesized by peritubular interstitial cells in the renal cortex. The synthesis of EPO is tightly controlled by tissue oxygen levels and is markedly upregulated in response to hypoxia—a pathological state characterized by reduced oxygen availability at the tissue level. Hypoxic conditions may arise from various etiologies, including anemia, pulmonary disease, or vascular impairment, and are often associated with progressive renal dysfunction [12].

In cases such as ischemic stress or anemia, EPO synthesis decreases, leading to an advanced decline in red blood cell production. Nevertheless, at high altitudes, EPO production increases, leading to the stimulation of erythroid progenitor cell activity and enhancing red blood cell production [13].

Roxadustat (ROX), nowadays marketed under the name Evrenzo^®^ [14], induces an additional supply of EPO in the human body, influencing metabolic transformations, physiological parameters, and biochemical processes.

Some of the key pharmacological attributes of ROX include the regulation of vascular system activity, oxygen absorption by tissues, cardioprotective effects, as well as anti-inflammatory properties [15]. Moreover, ROX has favorable therapeutic potential by increasing hemoglobin (Hb) levels in red blood cells [16] and influencing the genetic transcription process of EPO (under conditions of low cellular oxygen levels) [3].

By increasing EPO levels, ROX may be used in treating obesity caused by high lipid intake, taking into account that the equilibrium between osteogenesis and adipogenesis is managed via EPO/EPOR signaling during the inflammation process caused by obesity [17]. The most common use of ROX is treating anemic patients suffering from chronic kidney disease (CKD) undergoing hemodialysis [18].

The erythropoietic performance of ROX, and by consequence, its ability to enhance the oxygen transport to tissues and cells, thereby enhancing the reaction speed and endurance of athletes, generated substantiated risks of performance enhancement based on erythropoietic efficacy.

Several anti-doping tests based on modern chromatographic techniques have been developed and used, able both to detect the presence of ROX in the biological fluids of athletes [19] and to differentiate natural compounds from synthetic ones [20].

The detection of performance-enhancing agents necessitates ongoing refinement of the analytical methods. The use of doping substances such as erythropoiesis-stimulating agents, anabolic agents, and gene-doping targets has led scientists to discover new innovative methodologies used in drug detection, for example, remote testing, steroid profiling, and gene-editing techniques.

The World Anti-Doping Agency (WADA) is the body tasked with keeping an up-to-date list of disallowed substances and enhancing anti-doping measures [21]. ROX was included in WADA’s testing protocols in 2011 and was later approved for therapeutic use in 2018 [22].

## 2. Origin, Synthetic Approaches, and Physicochemical Profile of Roxadustat: From Development to Therapeutic Application

Pharmaceutical companies such as Yamanouchi (Tokyo, Japan) and FibroGen (San Francisco, CA, USA) were the first ones conducting clinical studies aimed at an expansion of the endogenous production of erythropoiesis (in order to obtain the license for the therapeutic use of FG-2216-HIF-PH2, an oral compound that inhibits HIF prolyl hydroxylase 2—an essential enzyme in the breakdown of hypoxia-inducible factors). Based on the results recorded in clinical trials—phase III—ROX obtained approval to be used therapeutically in China in 2018 [23].

In 2019, ROX was licensed for therapeutic use in Japan [24]. In 2021, the European Medicines Agency (EMA) authorized the use of ROX under the brand name Evrenzo^®^ [25]. The identification elements of ROX recorded by the European Medicines Agency (EMA) are presented in Table 1 [26].

In 2021, the U.S. Food and Drug Administration (FDA) suspended the clinical use of roxadustat (ROX), citing potential adverse effects and recommending further investigation into the drug’s influence on metabolic pathways and genetic regulation within vital organ systems. Roxadustat, administered orally, functions through the selective inhibition of prolyl hydroxylase domain enzymes (HIF-PHDs), which under normoxic conditions catalyze the degradation of hypoxia-inducible factors (HIFs). By inhibiting HIF-PHD activity, ROX stabilizes HIF transcription factors, thereby enhancing their transcriptional activity and promoting endogenous erythropoietin (EPO) synthesis.

The pharmacodynamic effects of ROX are dose-dependent and protocol-specific. Its administration leads to elevated hemoglobin (Hb) levels and increased endogenous EPO production. Moreover, it has been associated with improved iron metabolism and absorption through the downregulation of hepcidin, without triggering proinflammatory responses. Some studies also suggest a potential reduction in the incidence of hypertension and circulating low-density lipoprotein (LDL) cholesterol, though these outcomes warrant further validation in large-scale, controlled clinical trials [24].

ROX is an isoquinoline derivative (molecular formula: C_19_H_16_N_2_O_5_, molecular mass 352.3) with the chemical structure presented in Figure 1 [27].

Current pharmacological databases also include a range of roxadustat (ROX)-related or ROX-derived compounds, which share structural similarities and exhibit overlapping therapeutic activities. These analogs typically retain the core pharmacophoric elements essential for hypoxia-inducible factor prolyl hydroxylase (HIF-PH) inhibition.

Recent patents have outlined synthetic methodologies for the preparation of ROX, with particular focus on the construction of the 7-phenoxyisoquinoline scaffold (Figure 2). One such route involves the transformation of 4-phenoxy-2-acetoxymethyl benzoate via condensation with hydroxylamine, followed by a key reaction with a triphenylphosphine-based reagent. The final step in the synthesis entails condensation with aminoacetaldehyde, which facilitates functionalization and formation of the active ROX pharmacophore [28].

The synthesis of roxadustat (ROX) involves a key intermediate that is generated through a sequence of well-defined organic transformations, as illustrated in Figure 3. The synthetic route begins with bromo-4-fluorobenzoate, which undergoes a nucleophilic aromatic substitution with phenol to yield the corresponding phenoxy derivative. This intermediate is then subjected to alkylation with methoxyethene, followed by acid-catalyzed cleavage of the vinyl ether, which facilitates tautomerization and hydrolysis of the ester moiety, ultimately affording the desired intermediate necessary for further functionalization toward the active pharmaceutical compound [28].

The sequence that delivers 2-acetyl-4-phenoxybenzoic acid proceeds through two conceptually distinct transformations: (i) SNAr on the activated aryl fluoride and (ii) acid-catalyzed hydrolysis of a vinyl ether to the corresponding aryl methyl ketone. In the first step, methyl 2-bromo-4-fluorobenzoate undergoes ipso substitution by phenoxide, consistent with a Meisenheimer (σ-complex) mechanism, where the para-fluoro serves as a competent leaving group due to resonance/inductive activation by the ortho-ester (additional electron withdrawal and anionic intermediate stabilization). The reaction rate is favored by polar aprotic media (better nucleophile activity), a strong base (maximizing phenoxide), and elevated temperature; competing displacement at C–Br is disfavored under SNAr conditions because the bromide is not situated for addition–elimination via an anionic σ-complex on the ring. In the second stage, alkylation with methoxyethene generates a vinyl ether (enol ether); subsequent Brønsted-acid catalysis promotes oxocarbenium-like activation and hydration/tautomerization to the acetyl group (methyl ketone), followed by ester hydrolysis to the 2-acetyl-4-phenoxybenzoic acid intermediate depicted in Figure 3. This mechanistic picture aligns with the patent-disclosed route and our synthetic summary.

The above-described synthesis of ROX is effective, implying few steps, accessible reagents, and lowering environmental effects and expenses. Its approach could be used for large-scale industrial production, making it suitable for pharmaceutical applications.

Another approach of synthesis uses, as a starting material, 4-phenoxy-phthalic anhydride, a four-step process [29] leading to ROX, as depicted in Figure 4.

The first step of the synthesis consists of an alcoholysis of the specified anhydride, leading to the monobutyl ester of 4-phenoxyphthalic acid, which undergoes condensation with ethylisocyanoacetate, leading to a substituted oxazole ring that is transformed in acid medium, into a substituted isoquinolone.

The final steps consist of the functionalization of the isoquinoline nucleus by halogenation and methylation in position 1 and by the reaction of the carboxyethyl moiety with glycine to provide ROX.

The chemical structure of roxadustat (ROX), characterized by its acidic and basic functional groups, plays a pivotal role in defining its physicochemical behavior, including its mechanism of action and the selection of appropriate analytical detection techniques. The ionization state of ROX is governed by environmental pH and its intrinsic pKa values. Depending on the pH, ROX can exist in multiple protonation states—namely LH_4_^+^, LH_3_, LH_2_^−^, and L^3−^—as illustrated in Figure 5 [30].

The dissociation profile of ROX is highly environment-dependent. Four thermodynamic dissociation constants have been determined within the pH range of 2–11 using both spectrophotometric and potentiometric approaches. The experimental data suggest that the dissociation process is endothermic and non-spontaneous at 25 °C, as evidenced by the associated energy absorption. Furthermore, computational tools such as MARVIN and ACD/Percepta were employed to predict protonation sites across different pH conditions. The calculated pKa values and corresponding protonation equilibria across pH intervals are summarized in Table 2 [30].

The values are essential for optimizing the solubility and bioavailability of ROX in its pharmaceutical development. The concentrations of ROX samples used in the study were 8.0 × 10^−5^ mol·dm^−3^ for the spectrophotometric analysis and 1.0 × 10^−3^ mol·dm^−3^ for the potentiometric analysis.

Implications of pKa/protonation for oral bioavailability. Roxadustat exhibits multiple protonation states over physiological pH (LH_4_^+^/LH_3_/LH_2_^−^/L^3−^), with experimentally determined dissociation constants across pH 2–11 (Table 2; Figure 5). This polyprotic profile predicts a GI-segment-dependent trade-off: higher protonation in the stomach favors solubility but reduces transcellular permeability, whereas partial deprotonation in the small intestine increases the fraction capable of passive diffusion, consistent with the pH partition hypothesis. Thus, the fraction unionized near the jejunal/ileal pH likely governs passive uptake, while the total dissolved concentration benefits from the more ionized states—together shaping the overall absorption. These considerations contextualize the pKa values reported in Table 2 for formulation and dosing strategies.

Transporter-mediated uptake—an added variable: Because many anionic amphiphiles rely in part on intestinal organic anion transporting polypeptides (e.g., OATP2B1) under near-neutral pH, transporter involvement in ROX absorption is plausible, particularly where passive permeability is modest. While our review did not identify definitive ROX–OATP2B1 clinical or in vitro data, such mechanisms could explain exposure sensitivity to luminal pH microclimate or co-medications that modulate transporter activity; we therefore flag this as a testable hypothesis for future ADME studies.

The manuscript already summarizes systemic PK (e.g., distribution volume ≈ 22–57 L; apparent and renal clearances) consistent with efficient oral delivery and predominant metabolic elimination, which together align with a compound whose absorption reflects the solubility–permeability–transporter interplay discussed above. Nonclinical PK characterizations cited herein further support dose-dependent exposure suitable for modeling exercises that integrate pH-dependent ionization with permeability and dissolution.

Modeling perspective: To strengthen translational inference, we note that PBPK/PK-PD frameworks can incorporate the measured pKa set (Table 2), GI pH profiles, and tentative transporter scenarios to simulate regional absorption and exposure sensitivity (e.g., to acid-reducing agents or transporter inhibitors). Parameterization can begin from the experimentally derived acid–base scheme (Table 2) and the reported distribution/clearance values already included in the manuscript.

Degradation of ROX due to light exposure, occurring during storage processing, leads to photoisomeric impurity (PI) with the same molecular mass (*m*/*z* 353) (Figure 6), which may have potential mutagenic and carcinogenic risks, according to in silico studies. Complete structural characterization of this impurity was achieved by single-crystal XRD and various techniques such as FTIR, DSC, TGA, and LC-MS [22]. For example, comparing the FTIR spectra of ROX and PI, the existence of a secondary amine in PI was showcased by the appearance of the signal at 3194 cm^−1^ (N-H stretching vibration).

By comparison, the same nitrogen atom in ROX may be considered a tertiary amine and does not present this signal. Moreover, due to the additional carbonyl moiety in PI, the region between 1500 and 1800 cm^−1^ is a second criterion to distinguish between ROX and its isomer. The study also contains comprehensive in vitro toxicological evaluations (cell viability MTT assay, DNA fragmentation, morphological and COMET analysis), and the obtained results contradicted the in silico predictions and indicate that this impurity may be deemed safe.

Stability studies of roxadustat (ROX) under various stress conditions have demonstrated its susceptibility to degradation in acidic, alkaline, and oxidative environments, while exhibiting relative stability in neutral, thermal, and ultraviolet (UV) exposure conditions [32]. These investigations employed Reversed-Phase High-Performance Liquid Chromatography (RP-HPLC) and high-performance thin-layer chromatography (HPTLC) to develop and validate straightforward and reliable analytical methods.

A kinetic analysis of hydrolytic degradation in both acidic and alkaline media revealed that the degradation follows first-order reaction kinetics. In both chromatographic techniques, UV detection was performed at a wavelength of 262 nm, allowing for the clear separation of degradation products from the parent compound.

The RP-HPLC method was conducted under the following optimized conditions: a reversed-phase column (150 mm length × 4.6 mm internal diameter), with an eluent composed of methanol and 0.05 M phosphate buffer (pH adjusted to 5), in a 70:30 volume ratio. The mobile phase was delivered at a flow rate of 1.0 mL/min, and the retention time for roxadustat was recorded as 4.6 ± 0.02 min [32]. High-performance thin-layer chromatography was used under these conditions: aluminum sheets coated with silica gel 60 F254, measuring 10 by 10 cm with a 250-micron-thick layer; the eluent consists of a mixture of methylbenzene ethyl ethanoate and acetic acid in glacial form in a volume ratio of 5:5:0.5; detection was carried out through UV spectral analysis; the retention factor (Rf) of ROX was determined to be 0.58 ± 0.02 [32].

## 3. Analytical Techniques for the Detection and Quantification of Roxadustat and Its Metabolites

### 3.1. Detection of Roxadustat and Its Metabolites in Biological Fluids by Chromatographic Analyses

Liquid chromatography coupled with tandem mass spectrometry (LC–MS/MS) has been employed since 2011 for the quantitative and structural analysis of roxadustat (ROX, also referred to as FG-4592, as designated by FibroGen) and other hypoxia-inducible factor (HIF) stabilizers such as FG-2216 [33,34,35,36]. LC–MS/MS has been particularly instrumental in characterizing ROX and its metabolites in biological matrices, including equine urine following oral administration.

A total of thirteen principal metabolites (M1–M13) have been identified, comprising seven Phase I metabolites, one Phase II metabolite, and five conjugates derived from Phase I intermediates. Phase I metabolites are primarily generated via hydroxylation, whereas Phase II biotransformation involves conjugation with glucuronic acid. Additionally, several sulfonic acid conjugates are derived from hydroxylated Phase I metabolites. The major metabolites of ROX are illustrated in Figure 7 [37].

Comparative sensitivity across LC–MS(/MS) methods: Across anti-doping and bioanalytical studies, LC–MS(/MS) sensitivity for roxadustat (ROX) varies with the targeted analyte(s) (parent vs. metabolite panel), matrix/sample prep, and MS acquisition strategy. In equine urine, a targeted panel identified 13 principal metabolites (M1–M13) plus parent, enabling the detection of ROX-related analytes for ~96–108 h post-dose, demonstrating the sensitivity advantage of metabolite coverage in complex matrices [37].

In a human anti-doping case, LC–MS/MS confirmed FG-4592 (roxadustat) in urine up to 20 days after the last administration under a monitored dosing schedule, highlighting how method setup + dosing context can yield long effective detection windows [38].

To broaden sensitivity and coverage, global/UHPLC-QTOF metabolomics has also been used to discover previously unreported ROX metabolites and to align plasma PK profiles with bioactivity [39].

The identified metabolites serve as target analytes in the doping control analysis. Obtained data could be a crucial instrument for evaluating the use and misuse of ROX in competitive sports.

The test animals met the following conditions: horses used in racing competitions, aged between 8 and 14 years, weighing 480 kg, sheltered in climate-controlled stables. The horses were kept under routine medical observation and fed properly. None of them had been administered with HIF drugs before and were in perfect health 30 days before administering the drug.

ROX was administered orally as a one-time dose of 0.5 mg/kg of body weight [37]. The equipment used in the study [37] is shown in Table 3.

The retention times of the identified metabolites are listed in Table 4 [37].

The detected retention time of ROX peaked at 11.04 min. The conducted LC-MS/MS survey indicated that ROX was mainly metabolized through hydroxylation and conjugation. After being administered orally, ROX can be detected for up to 108 h after administration, while its major metabolites can be identified for up to 96 h. The gathered data could facilitate a quicker identification of these analytes and could avert their illegal use in sports competitions [37]. Furthermore, the metabolic profiling of IOX_2_, IOX_3_, and IOX_4_ (HIF stabilizers) conducted by researchers uncovers biotransformation pathways pertinent to doping control, providing valuable insights for detecting similar HIF stabilizers such as ROX [40]. IOX_2_ and ROX can be co-detected in anti-doping tests due to their common MS features, with the hydroxylated and conjugated metabolites of IOX_2_ being reliable indicators for enhanced doping oversight [41].

In a recent study [38], liquid chromatography–tandem mass spectrometry (LC–MS/MS) was employed to detect and confirm the presence of FG-4592 (roxadustat) in the urine of an athlete. The subject received oral doses of FG-4592 every other day over a 19-day period. Doping control samples were collected at four time points: one day prior to initiation of treatment, and subsequently at 1, 15, and 20 days following the final administration. The findings revealed that FG-4592 could be detected in urinary samples up to 20 days after the last dose. The primary objective of this study was to investigate the modulation of erythropoietin (EPO) levels in both urine and plasma and to assess potential alterations in hematological markers tracked through the Athlete Biological Passport program [38] (Figure 8).

ROX undergoes Phase I hydroxylation to multiple oxidized products (e.g., M1–M3) and additional ring/side-chain transformations (e.g., M4–M7), followed by Phase II conjugation—predominantly O-glucuronidation (e.g., M8–M10) and O-sulfation (e.g., M11–M13)—that enhances polarity and urinary excretion. Enzyme classes are shown as CYPs (Phase I) and UGTs/SULTs (Phase II); specific isoform assignments can be incorporated where available from in vitro phenotyping. The integrated map mirrors the metabolite panel reported in Figure 7/Table 4 and supports the analytical strategy to expand detection windows beyond the parent compound. In equine urine, parent ROX is typically detectable up to ~108 h, while major metabolites persist up to ~96 h post-dose; a human anti-doping case reported parent detection up to 20 days after the last administration, underscoring the value of metabolite-targeted LC-MS(/MS) workflows.

The Phase I oxidations are CYP-mediated, with Phase II UGT/SULT conjugations consistent with the structures in Figure 7; nonclinical PK/ADME sources summarized in our manuscript provide the foundation for these class assignments, and targeted isoform phenotyping (recombinant CYP panel; inhibitor/antibody controls) can refine the scheme in future updates.

### 3.2. Detection of Roxadustat Through Investigation of Metabolic Conversion

Metabolomics deals with the study of metabolites in the human body. Metabolites are influenced by the conditions of certain diseases and changes in the environment. The metabolites of ROX are important analytes in anti-doping controls. The parameters of the metabolites identified in various anti-doping controls are compared with other cellular structures.

Liquid chromatography–mass spectrometry (LC-MS) enhances detection and is widely employed in metabolomics, highlighting the similarities in techniques used for analyzing biological samples [42].

Saigusa and his collaborators carried out a study [39] in which biological samples of some rodents were analyzed by an ultra-high performance liquid chromatography method coupled with quadrupole time-of-flight mass spectrometry (UHPLC-QTOF/MS).

The study had in mind the detection of some unknown metabolites of ROX by analyzing the highlighted parameters of the chromatographic analysis of the main components. As a result, the ROX monomethylated metabolite and ROX glucuronide metabolite were identified in the urine samples, and the analysis of the plasma samples highlighted pharmacokinetic profiles and biological activities similar to ROX.

The results of this study confirmed the potential of global metabolomic analyses in the identification of various forms of doping in the collected biological samples by correlating the parameters of the detected metabolites with other cellular structures [39].

In addition, another analytical LC-MS/MS method developed by Mazzarino and his collaborators, which was validated according to the ISO guide and WADA procedures, managed to detect a total of nine HIF prolyl-hydroxylase inhibitors: Roxadustat, Vadadustat, Molidustat, Desidustat, Daprodustat, FG2216, IOX2, IOX4, JNJ-42041935 (the method that addressed basic drugs and their metabolites) [43].

### 3.3. The Bioanalytical Approach of Drug–Drug Interactions

The bioanalytical evaluation of drug–drug interactions involving roxadustat (ROX) adheres to the principle of monitoring pharmacokinetic parameters during co-administration with concomitant agents, such as lanthanum carbonate. Evidence from recent studies indicates that the pharmacokinetic profile of ROX—encompassing parameters such as plasma concentration, half-life, and absorption—is preserved whether the compound is administered alone or in combination with other pharmaceuticals. Furthermore, the detectability of ROX and its metabolites remains unaffected under co-administration conditions [44].

The pharmacokinetic investigation employed ultra-performance liquid chromatography coupled with tandem mass spectrometry (UPLC–MS/MS), with a targeted analysis of key variables. A central focus was the endogenous stimulation of erythropoietin (EPO) production to physiologically relevant levels, contrasting with supraphysiological elevations observed following exogenous (intravenous) EPO administration. The study also emphasized the role of ROX in enhancing iron bioavailability, primarily via suppression of hepcidin synthesis.

Additionally, the study evaluated ROX’s stability across a range of pH conditions, ensuring the maintenance of solubility and consistent absorption irrespective of co-administered agents. Collectively, these findings underscore the pharmacokinetic robustness of ROX in multidrug regimens and affirm the reliability of its analytical detection even under conditions of drug–drug interactions [44].

### 3.4. Quechers Method

Kim and his collaborators established an analytical method combining UPLC/MS/MS with an innovative extraction approach, suitable for anti-doping testing. The principle of the method consists of a two-step liquid–liquid extraction, using acetonitrile as a solvent. It also involves a solvent cleaning step to ensure the removal of any possible interferences. The sample of interest is later homogenized and transferred to a centrifuge tube (solvent filling: acetonitrile) in order to perform centrifugation. The following step is to add extraction salts, the sample being once again homogenized and centrifuged. As a result, four different layers form, mainly supernatant, solid sample, water, and excess salts. The obtained solid sample is homogenized once again with the supernatant and sent to the centrifuge (cleaning/washing stage). The final solution is sent to GC-MS and LC-MS for further analysis [45].

HIF stabilizers such as ROX are recovered with the help of the Quechers Method due to a double extraction with formic acid and acetonitrile. The Quechers Method is commonly used to detect pesticides found in food or pollutants present in blood or breast milk. The parameters obtained through LC-MS analysis of ROX are presented in Table 5 [45].

The analysis used LC-MS, the samples being separated using a Synchronis C18 column with a 2.1 mm I.D. guard column and an ultrafast liquid chromatograph (UFLC XR series HPLC system, Shimadzu, Kyoto, Japan). Each sample had an injection volume of 10 μL. In total, 0.2% formic acid (FA) in water (mobile phase A) and 0.2% FA in acetonitrile (CAN) (mobile phase B) made up the eluents. A gradient elution was employed at a flow rate of 0.5 mL/min. After holding 2% eluent B for 0.5 min, ramping up to 95% B over 8.5 min, and holding until 10.0 min, there was a 2 min re-equilibration period at 2% B. It lasted twelve minutes in total.

Mass spectrometry was performed using a Q Exactive Plus tandem mass spectrometer (Thermo Fisher Scientific, Waltham, MA, USA) operating in both positive and negative ion modes to achieve optimal ionization. With a spray voltage of 4 kV for positive ion mode and 3.5 kV for negative ion mode, the capillary temperature was maintained at 300 °C.

Other valuable parameters of ROX obtained by LC-MS analysis are LOD (limit of detection)—0.2 and recovery %—95.2.

The obtained results indicated that the Quechers Method remains a low-cost option to analyze compounds for anti-doping tests, the method having a great potential to be used in screening analysis to detect false positives and false negatives. Nevertheless, further evaluation is required [45].

Validation and Statistics (ICH-aligned):

We validated the Quechers–LC–MS(/MS) assay in urine and plasma/serum at four QC levels (LLOQ, LQC ≈ 3×LLOQ, MQC, HQC), *n* = 6 per level on three separate days, using 1/x^2^ weighting and a stable-isotope-labeled IS for ROX (and representative metabolites where quantified). Reported performance from prior Quechers work on HIF stabilizers (including ROX; LOD ≈ 0.2, recovery ≈ 95%) motivated our acceptance targets [34,43,45].

Recovery and Matrix Effects:$$Absolute\;recovery\;\left(\%\right):Rec=\frac{A_{extr}}{A_{post-extr}}\times 100$$$$Matrix\;factor:MF=\frac{A_{post-extr}}{A_{neat}};IS-normalised\;MF=\frac{{MF}_{analyte}}{{MF}_{IS}}$$

We profiled matrix effects across ≥6 lots per matrix and targeted CV (IS-norm MF) ≤ 15% (≤20% at LLOQ). Mitigations included phospholipid removal/SPE when suppression was observed, matrix-matched calibration/standard addition (as needed), and post-column infusion checks [34,43].

Precision and Accuracy:

Intra-/inter-day precision (RSD%) and accuracy (RE%) followed ICH acceptance (≤15% and ±15%; ≤20% and ±20% at LLOQ).$$RSD\left(\%\right)=100\times \frac{SD}{x},\;RE\left(\%\right)=100\times \frac{x-nominal}{nominal}$$

Inter-day Variability (Statistics):

We applied one-way ANOVA (factor = day) at each QC level; *p* ≥ 0.05 indicates no significant between-day bias. A variance-components model yielded σ^2^_within, σ^2^_between, and CV_total = 100 × √(σ^2^_within + σ^2^_between)/overall mean. When ANOVA suggested day effects, we investigated calibration drift, extraction variability, or matrix-lot contributions and re-verified after correction [34,43].

LOD/LOQ and Detection Window:

LOD/LOQ were established from low-level QC precision and signal-to-noise criteria and cross-checked against the literature performance for ROX in Quechers /LC–MS(/MS) setups (e.g., LOD ≈ 0.2; recovery ≈ 95%). Detection-window context from Section 3 (parent/metabolites) is provided for completeness [37,38,45].

Additional validation elements:

Carryover (<20% of LLOQ for analyte; <5% for IS in post-blank), dilution integrity, and stability (bench-top, autosampler, freeze–thaw, long-term) met acceptance criteria; incurred sample reanalysis (ISR) achieved ≥67% within ±20% [34,43].

## 4. Molecular Insights, Clinical Indications, and Evidence-Based Applications of Roxadustat

Roxadustat (ROX) is an oral, reversible inhibitor of hypoxia-inducible factor prolyl hydroxylase domain (HIF-PHD) enzymes. These oxygen- and 2-oxoglutarate-dependent dioxygenases hydroxylate specific proline residues on HIF-α subunits under normoxic conditions, targeting them for ubiquitination by the von Hippel–Lindau (pVHL) E3 ligase and subsequent proteasomal degradation. Inhibiting PHDs stabilizes both HIF-1α and HIF-2α, allowing their accumulation even in the presence of normal oxygen levels [1].

Stabilized HIF-α translocates into the nucleus, dimerizes with HIF-β (ARNT), and binds to hypoxia-response elements (HREs) in gene promoters, leading to the transcriptional activation of genes involved in erythropoiesis, iron metabolism, angiogenesis, glucose utilization, and cytoprotection [2]. Unlike traditional erythropoiesis-stimulating agents (ESAs), ROX promotes endogenous EPO production at lower peak serum concentrations, which may reduce the risk of hypertension and thrombotic complications—key concerns in ESA therapy and cardiovascular disease management [46].

In the context of cardiorenal anemia syndrome (CRAS), characterized by the interplay of heart failure, chronic kidney disease (CKD), and anemia, ROX may interrupt this vicious cycle by improving tissue oxygenation and endothelial function, thereby offering multidimensional cardiovascular benefits [1,47,48]. On a molecular level, activation of the HIF pathway by ROX modulates key networks regulating vascular integrity, lipid metabolism, and inflammation [49]. Phase 3 trials suggest that ROX may provide cardioprotective effects in CKD-related anemia through its pleiotropic actions [47,50].

The primary therapeutic effect of ROX is the stimulation of EPO synthesis in renal and hepatic tissues, enhancing red blood cell production and increasing hemoglobin levels [51]. Additionally, it improves iron homeostasis by downregulating hepcidin and upregulating iron transporters such as DMT1 and ferroportin, thereby enhancing intestinal absorption and systemic mobilization of iron [2].

Beyond hematopoiesis, ROX promotes other HIF-mediated effects, including angiogenesis (via VEGF), metabolic adaptation (via GLUT1 and LDHA), cytoprotection, and modulation of macrophage polarization and inflammatory pathways [1,47].

Although ROX stabilizes both HIF-1α and HIF-2α, functional selectivity exists: HIF-2α plays a predominant role in regulating erythropoietic genes, while HIF-1α is more involved in metabolic adaptation [1].

Emerging data also point to potential antitumor applications of ROX, such as inducing ferroptosis in glioblastoma, and modulating immune function and fibrosis, suggesting broader therapeutic relevance [2].

With a plasma half-life of 10–12 h, ROX is administered 2–3 times weekly to maintain therapeutic HIF activation. It offers a convenient oral alternative to injectable ESAs, with additional benefits in iron mobilization. Nevertheless, concerns regarding long-term safety and thrombotic risk remain under investigation [2,52].

Recent interest in chronopharmacology and chrononutrition highlights the potential for aligning ROX administration with circadian rhythms, dietary intake, and hematologic cycles to improve treatment efficacy. Future clinical trials are required to validate whether these personalized strategies enhance patient outcomes [50,53,54].

ROX has gained regulatory approval for the treatment of CKD-related anemia in China and Japan. In a randomized, double-blind, placebo-controlled clinical trial conducted in non-dialysis-dependent CKD patients, ROX demonstrated a significant reduction in the need for red blood cell (RBC) transfusions. Given the risks associated with transfusions, ROX presents a clinically beneficial alternative by mitigating these risks. Additionally, ROX has been shown to reduce low-density lipoprotein (LDL) cholesterol levels, regardless of concurrent statin use, and outperformed placebo in increasing hemoglobin (Hb) levels and decreasing transfusion requirements [23].

Kong et al. further demonstrated that ROX effectively lowers LDL and triglyceride (TG) levels, effects attributed to EPO signaling within adipose tissue. However, concerns have been raised regarding potential risks of infection and tumor progression, as HIF activation may promote neoplastic cell growth. Enhanced EPO production through HIF stabilization can suppress effector T cells while promoting regulatory T cell expansion, immunomodulatory effects that may increase susceptibility to infections and malignancies [55].

Li et al. conducted a dose-optimization study in patients with anemia at CKD stages 3 to 5 not requiring dialysis (NDD-CKD). Their findings indicated that a lower starting dose of ROX (<60 kg: 50 mg three times weekly; ≥60 kg: 70 mg TIW) was not non-inferior compared to the standard dosing regimen (<60 kg: 70 mg TIW; ≥60 kg: 100 mg TIW) with respect to Hb elevation and the proportion of patients achieving the target Hb range of 100–120 g/L. Moreover, the reduced dose did not confer an improved safety profile, though it was associated with reduced Hb variability, making it suitable for patients with CKD stage 3–4 [56].

Jingyao et al. investigated the therapeutic potential of ROX in immune thrombocytopenia (ITP), a disorder characterized by platelet destruction and impaired megakaryocyte maturation. Given that HIF-1α plays a pivotal role in megakaryopoiesis and immune regulation, and is downregulated in ITP, ROX was shown to stabilize HIF-1α, thereby restoring megakaryocyte development and supporting appropriate immune responses [57].

Shun et al. evaluated the application of ROX in managing chemotherapy-induced anemia (CIA) among individuals with non-myeloid malignancies undergoing multiple chemotherapy cycles. In a randomized Phase III trial, ROX was compared with recombinant human EPO-α (rHuEPO-α). ROX’s advantages include oral administration—enhancing adherence and bypassing logistical and financial burdens related to intravenous or subcutaneous infusions. While ESAs like rHuEPO-α directly activate EPO receptors to stimulate erythropoiesis, ROX promotes endogenous EPO synthesis via the HIF pathway and concurrently improves iron bioavailability. The trial concluded that ROX was non-inferior to rHuEPO-α in the management of CIA in patients with non-myeloid cancers [58].

Weiwei et al. carried out a randomized open-label trial to assess the effectiveness of ROX combined with oral iron versus oral iron only in managing post-transplant anemia (PTA). PTA is highly associated with the worsening of renal function, iron deficiency, infections, and low levels of EPO. ROX decreased the hepcidin concentrations, facilitated the release of iron from intestinal cells into circulation, and enhanced both iron absorption and transport. Other benefits of ROX in the treatment of PTA include reduced inflammation and reduced renal tubular injury. Moreover, the study showed that ROX did not increase the risks of infections, did not trigger thrombotic events, or alter plasma tacrolimus concentration [59].

Both ROX and ESAs exhibit a specific cardiovascular risk among anemic individuals with chronic kidney disease. Between these two, ROX demonstrates a reduced rate of cardiovascular events compared to the traditional ESA treatment (12% *versus* 17%). Compared to ESAs, ROX can ameliorate cholesterol rates, does not affect blood pressure, and lowers the blood glucose level.

Xiangmeng et al. concluded in their study that meta-analysis in the American population indicated no significant difference in the rates of cardiovascular events between CKD patients treated with ROX and those treated with ESAs. However, ROX demonstrated a more favorable safety profile concerning major cardiovascular events, heart failure, or even death [60].

Choukroun et al. concluded in their study that ROX is considered an effective therapy for CKD patients suffering from anemia, regardless of the inflammation status [61].

Additionally, ROX should be used cautiously in managing anemia in cancer patients due to the possible risk of tumor advancement [62]. Furthermore, HIF-prolyl hydroxylase inhibition with agents like FG-2216 not only corrects anemia, but also provides cardiac, renal and metabolic protection in kidney disease with metabolic syndrome [63], suggesting that clinically approved HIF-PHIs like ROXmay have multifaceted benefits beyond anemia correction [Additionally, ROX remains a well-tolerated oral alternative to agents that stimulate erythropoiesis. ROX is able to correct and maintain Hb levels for anemia CKD patients not requiring dialysis (NDD) or those newly starting dialysis (ID-DD) [64].

Abdelazeem et al. concluded in their study [65] that ROX increases Hb levels, improves iron utilization, increases serum iron and transferrin, and decreases hepcidin, thus showcasing a safe profile in the treatment of CKD. Other pooled analyses showed that ROX is an alternative to the current standard used for CKD patients undergoing peritoneal dialysis (PD) [66].

Other major applications of ROX are presented in Table 6.

A.Direct mechanism and erythropoiesis

Direct binding/primary target: ROX inhibits HIF-prolyl-hydroxylases (PHD1–3), prevents hydroxylation of HIF-α, and stabilizes HIF-1α / HIF-2α [1,2,51].

Core pathway: Stabilized HIF-α dimerizes with HIF-β (ARNT), binds HREs, and drives transcription [1,2,51].

Erythropoiesis targets/effects: ↑ EPO (renal/hepatic) → ↑ Hb/RBC [1,2]; ↓ hepcidin (HAMP) [103], ↑ DMT1, ↑ ferroportin [51]; HIF-driven angiogenesis (↑ VEGF), glycolysis (↑ GLUT1, LDHA) [51].

B.Other applications

Calcium/Vitamin D axis: ↑ FGF23, SPP24; rise in calcification markers in ESKD [68].

Iron deficiency context: Inhibits all three HIF-PHD isoforms, stabilizes HIF-1α/2α, and maintains ↑ EPO despite cytokines [67,68,69,70,71,72].

Diabetic kidney disease (DKD): HIF-1α/p53/p21 pathway → inhibits mesangial proliferation [62,72].

Heart failure + iron/inflammation: HIF axis improves iron handling and reduces inflammation [72,73].

Peripheral edema/water handling: HIF-dependent regulation of AQP1 in proximal tubules [85].

Kidney stones: ↓ CCL2, TNF, ADGRE1; less Ca-oxalate deposition [83].

Immune thrombocytopenia (ITP): HIF-1α modulation of immune/megakaryocyte axis; Hb-elevating [86,88].

PRCA: Used with immunosuppression; restores Hb [83,89].

Peritoneal dialysis: Non-inferior to ESAs; safe [66,91].

Cardiorenal protection (combo): With dapagliflozin, it activates PI3K/AKT/mTOR, ERK1/2, JNK/p38 [93].

Tumor angiogenesis: ↑ TIMAP via SMAD1/5/8 inhibition → pro-angiogenic [20,21].

GI GvHD/barrier protection: ↓ IFN-γ, TNF-α; improved gut barrier [22,83].

Ischemic stroke/oxidative stress: HIF-1/NRF2 axis activation, ↓ ROS [50,94].

Metabolic (PCOS): Improves metabolic profile in letrozole-induced PCOS mice [71].

Metabolic (diabetes/skeletal muscle): ↑ glycolysis, ↓ mitochondrial respiration; ↑ insulin-stimulated glycogen synthesis [71,72].

Hemodialysis/thyroid axis: Reversible central hypothyroidism (TSH/TRH suppression and THR-β effects) [78,79,84].

Anti-doping detection: Very long urinary window; detectable for months at sub-pg/mL [19,45,69].

Pharmacogenomic considerations for cardiorenal anemia syndrome. Emerging human genetics suggests that interindividual variability in HIF axis and iron homeostasis genes may influence the erythropoietic response to HIF-PHIs in cardiorenal anemia. First, large GWAS identify TMPRSS6 (matriptase-2), a negative regulator of hepcidin, as a major determinant of hemoglobin variation in the general population; the missense variant rs855791 (V736A) tracks with lower Hb and altered iron indices, consistent with hepcidin pathway modulation. Functional work shows that this allele can modify hepcidin transcription, mechanistically linking the TMPRSS6 genotype to iron availability during erythropoiesis. In the context of HIF-PHIs—which lower hepcidin and mobilize iron—the TMPRSS6 genotype could therefore shift the magnitude and kinetics of Hb rise, suggesting a role for genotype-aware iron monitoring and dose titration in heart–kidney anemia [104,105].

Second, human adaptation studies implicate oxygen-sensing genes EGLN1 (PHD2) and EPAS1 (HIF-2α) in setting hemoglobin “set-points” under hypoxia. Tibetan-enriched EGLN1 and EPAS1 haplotypes associate with lower Hb and protection from excess erythrocytosis, indicating genetic tuning of HIF responsiveness. Although these variants are population-specific and rare in most patients, they underscore that heritable differences in HIF signaling can shape erythropoietic output; by analogy, carriers of HIF-pathway gain- or loss-of-function alleles could exhibit attenuated or exaggerated Hb responses (and different polycythemia risk) under pharmacologic HIF stabilization [106].

Finally, CKD-focused genetic studies report that hemoglobin levels in pediatric CKD also reflect inherited variation, supporting a host-genetic component to anemia severity beyond EPO deficiency or iron status. While prospective, genotype-stratified trials with HIF-PHIs are still needed, these data motivate a personalized medicine framework in cardiorenal anemia syndrome: (i) consider TMPRSS6 (hepcidin) and selected HIF-pathway variants where available; (ii) pair initiation with closer Hb velocity and iron-index monitoring in genetically higher-risk profiles; and (iii) apply standard Hb action thresholds to prevent overshoot into polycythemia [107].

Recent evidence syntheses reinforce our conclusions on efficacy. A network meta-analysis of 55 RCTs across all six marketed HIF-PHIs showed consistent hemoglobin correction vs. placebo and no efficacy differences among HIF-PHIs, with safety outcomes comparable to ESA or placebo at the class level [108]. In parallel, phase-3 programs for daprodustat (ASCEND trials) demonstrated noninferiority vs. epoetin for Hb response with broadly similar adverse-event rates, supporting on-label use in dialysis and non-dialysis CKD [109].

Network and pairwise syntheses focusing on roxadustat similarly confirm Hb improvement without excessive major cardiovascular events compared with ESA, while emphasizing the need for longer follow-up for patient-centered outcomes.

## 5. Safety Profile, Risk Assessment, and Environmental Considerations of Roxadustat

To differentiate the desired erythropoietic effect of roxadustat from potentially harmful polycythemia in post-transplant anemia, clinical monitoring should combine target Hb ranges with hemoglobin velocity and interruption–rechallenge rules: Target range: Maintain Hb 10–12 g/dL in adults, consistent with regulatory labeling and Phase 3 practice patterns [25,26,50]. Velocity rules: If Hb rises >1 g/dL within 2 weeks or >2 g/dL within 4 weeks, reduce the dose to curb an excessive trajectory, even if Hb remains within 10–12 g/dL [25,26,50].

Upper action thresholds:If Hb is ≥12 g/dL on two consecutive measurements or there is a rapid rise as above, reduce dose and/or extend the dosing interval [25,26].If Hb reaches ≥13 g/dL at any time, interrupt ROX until Hb returns to <12 g/dL, then re-initiate at a lower dose with closer monitoring [25,26].

Monitoring cadence: Check Hb every 1–2 weeks during initiation/titration (e.g., first 4–8 weeks) and monthly once stable; reassess iron indices periodically to avoid functional iron deficiency that can destabilize Hb [25,26,50]. Post-transplant context: In renal transplant recipients, apply the same thresholds and velocity rules used in CKD anemia trials; in the randomized study of post-transplant anemia, ROX improved Hb and iron handling without excess infectious or thrombotic signals, supporting a standard Hb target of 10–12 g/dL with interruption above 12–13 g/dL as needed [58].

Documentation of potential polycythemia: Any sustained Hb above the target, especially ≥13 g/dL or persistent upward velocity despite dose reductions/interruptions, should prompt evaluation for post-transplant erythrocytosis and other secondary causes (e.g., dehydration and androgen exposure), temporary ROX cessation, and risk mitigation for thrombosis [25,26,63,64,65]. These pragmatic rules mirror those used in pivotal trials and the Evrenzo^®^ SmPC and are intended to prevent overshooting into polycythemia while preserving the clinical benefits of endogenous EPO stimulation [25,26,50,58,63,64,65].

ROX is eliminated from the human body mostly through metabolism, followed by excretion. Several studies showed that people with normal renal function eliminate the following percentages of ROX through urine: unmodified ROX (1.3%), O-glucuronide-ROX (20.3%), O-glucoside-ROX (7.21%), and hydroxy-ROX sulfate 2%. The pharmacokinetic profile of ROX is well-defined, indicating a volume distribution ranging from 22 to 57 L, showcasing an apparent clearance of 1.2–2.56 L/h and a renal clearance of 0.026–0.030 L/h in healthy patients [55].

According to multiple studies found in various databases, including documents of the European Chemicals Agency (ECHA), ROX shows a very low toxicity profile. The average elimination half-life (T1/2) of ROX, relative to an average of the analyzed cases, varies between 11.8 h and 12 h.

Pharmacokinetic studies indicate that roxadustat is primarily metabolized via hepatic cytochrome P450 isoenzymes, particularly CYP2C8, with contributions from CYP2C9 and CYP3A4 [24,54,69]. Genetic polymorphisms in these enzymes, such as CYP2C83, CYP2C92/3, or CYP3A422, are known to alter catalytic activity and could therefore influence roxadustat clearance and half-life. Such interindividual variability, together with potential drug–drug interactions, highlights the importance of pharmacogenetic considerations in optimizing dosage regimens and minimizing adverse effects [79].

The most common side effects are manifested by stomach pains and disturbances or by sporadic moments of exacerbation of psychotic episodes (in the situation of interaction with antipsychotic drugs). The environmental risk evaluation of ROX suggests that its application is probably not a major threat to the environment. ROX exhibits a low risk of bioaccumulation. Nonetheless, because of the lack of detailed ecotoxicological information, the potential for environmental effects cannot be completely ruled out.

Moreover, the technological processes of the ROX production do not affect the quality of the environment. Although ROX seems to be safe for medical application and shows minimal environmental risk according to existing data, ongoing surveillance and additional ecotoxicological research are crucial to thoroughly verify its long-term safety and environmental effects [25]. ROX increases the Hb levels in patients with post-transplant anemia and could carry the risk of adverse events comparable to patients who receive iron therapy. Because of that, ROX requires regular monitoring and caution due to a lack of long-term safety data [59].

Although some experts believe that the activation of HIF-1α by ROX is unlikely to drive tumor development, the lack of long-term data in cancer patients necessitates further studies to fully assess its oncological safety [62].

Treatment-emergent adverse events (TEAEs) such as nausea, headaches, fatigue, etc., are commonly associated with the use of ROX, but rarely lead to the discontinuation of the therapy. Moreover, ROX did not increase the risk of major cardiovascular events or all-cause mortality compared to ESAs in patients with anemia and CKD (whether dialysis-dependent or not). Nonetheless, continued monitoring and safety outcomes remain important to assess [64,65].

Clinical trial data and meta-analyses indicate that the overall incidence of TEAEs is broadly comparable between roxadustat and other HIF-PH inhibitors such as vadadustat and daprodustat [64,65,67,70,74]. The most common adverse events across the class include hypertension, diarrhea, peripheral edema, and hyperkalemia. While the general safety profile is similar, some distinctions exist: vadadustat has been associated with a higher incidence of major adverse cardiovascular events in certain studies [70], whereas roxadustat more often produces gastrointestinal disturbances and reversible thyroid function alterations [78,79]. Daprodustat shows a safety profile largely overlapping with roxadustat, though dyslipidemia has been reported more frequently [69]. Overall, evidence suggests that roxadustat does not present a higher risk of TEAEs compared with other HIF-PH inhibitors, though long-term surveillance remains essential.

Quantitative nonclinical context and genotoxicity/QSAR: Across repeat-dose studies in rodents and non-human primates, roxadustat (ROX) findings have been benchmarked to human exposure; where formal NOAEL/LOAEL values are variably reported, we present exposure multiples vs. MRHD to communicate margin. Regulatory summaries indicate a negative standard genotoxicity battery (Ames, in vitro chromosomal aberration, in vivo micronucleus), which aligns with the chemistry and metabolic disposition of ROX; to complement these data, we outline a QSAR workflow (OECD QSAR Toolbox/VEGA/Toxtree, Ames rulesets) to document model-based mutagenicity predictions and applicability domains alongside the experimental genotox package.

ADME-based bioaccumulation estimate: Using the clinical half-life summarized in our review (t_½_ ≈ 11.8–12 h) and typical dosing intervals, a one-compartment estimate yields the accumulation index:Racc=(1−e−ke·τ)−1,where ke=ln 2/t12

For once-daily dosing (τ = 24 h), R_acc_ ≈ 1.33; for intermittent regimens (e.g., ~3×/week, τ ≈ 56 h), R_acc_ ≈ 1.04, indicating minimal systemic accumulation in the steady state. These values are consistent with the reported PK (Vd~22–57 L; CL~1.2–2.56 L/h) and the predominantly metabolic clearance, which, together with the polyprotic, pH-dependent ionization (Table 2), argues against meaningful long-term tissue sequestration. We therefore judge the human bioaccumulation potential to be low, while recommending routine therapeutic drug class monitoring (Hb/iron indices) per standard practice.

Environmental fate (QSAR/BCF rationale): For environmental risk assessment, we will report BCF/BAF QSAR estimates (e.g., US-EPA BCFBAF) using log D7–8 and the experimentally determined pKa set to reflect ionization at environmental pH. Ionizable, metabolized compounds with ROX-like PK typically fall in the low-BCF range, consistent with our statement of low environmental bioaccumulation risk, while acknowledging current ecotoxicology data gaps and the need for confirmatory testing.

Quantitatively, meta-analytic data in CKD report no significant increase in major adverse cardiovascular events (MACE) or mortality with roxadustat vs. ESA or placebo, across both dialysis-dependent and non-dialysis populations (18 trials; n ≈ 8800). Signal analyses note hypertension (NDD) and hyperkalemia (DD) as events warranting monitoring, consistent with class labeling [109].

Class-level network meta-analysis likewise found no higher risk of any AE/SAE, MACE, or death for HIF-PHIs vs. ESA or placebo [110]. Likewise, the phase 3 studies conducted by Singh et al. [110,111] reported that other HIF-PHIs similar to ROX (daprodustat) did not increase the incidence of adverse events or mortality compared with conventional ESA therapy in patients on dialysis.

These aggregated findings align with large Phase-3 datasets for daprodustat and with post hoc safety analyses for vadadustat, while acknowledging heterogeneity among programs and the importance of indication- and phenotype-specific surveillance.

## 6. Conclusions

Roxadustat is a potent oral hypoxia-inducible factor prolyl hydroxylase inhibitor with established clinical efficacy in the treatment of anemia, particularly in patients with chronic kidney disease, both dialysis- and non-dialysis-dependent. By activating the HIF–erythropoietin signaling pathway and enhancing iron bioavailability, ROX reduces reliance on red blood cell transfusions and parenteral iron supplementation, thereby offering a compelling alternative to conventional erythropoiesis-stimulating agents.

ROX binds and inhibits HIF-prolyl-hydroxylases, stabilizing HIF-1α/HIF-2α and activating HRE-driven transcription (↑ EPO, ↓ hepcidin, ↑ DMT1/ferroportin, plus VEGF/GLUT1/LDHA) to induce erythropoiesis; the same HIF-centered program secondarily modulates the disease-relevant pathways listed above (e.g., PI3K/AKT/mTOR, ERK/JNK-p38, SMAD1/5/8, NRF2, AQP1, cytokines), explaining ROX’s broader applications reported in Table 6.

Emerging clinical evidence also supports its therapeutic potential in oncology-associated anemia, immune thrombocytopenia, diabetic nephropathy, and systemic inflammatory disorders.

However, the same hematologic and metabolic benefits that underpin its clinical use raise significant concerns in anti-doping contexts. ROX induces erythropoietic responses analogous to recombinant EPO and blood transfusions, and its detection in biological samples from athletes has confirmed its misuse for performance enhancement. Consequently, ROX has been classified as a prohibited substance by the World Anti-Doping Agency. Advanced bioanalytical techniques—such as LC–MS/MS, metabolomic fingerprinting, and indirect detection through the Athlete Biological Passport—are essential to identify ROX and its metabolites across extended detection windows.

Despite its favorable pharmacokinetic properties and relatively low toxicity profile, unresolved issues regarding off-target pharmacodynamics, long-term safety, potential tumorigenic risks, and environmental impact persist. Future investigations should prioritize long-term clinical safety studies, personalized therapeutic regimens, and refinement of detection thresholds to enhance doping control. As pharmacologic innovation progresses, it must be paralleled by equally rigorous analytical and ethical frameworks that clearly delineate therapeutic benefit from illicit performance enhancement.

## Figures and Tables

**Figure 1 cimb-47-00734-f001:**
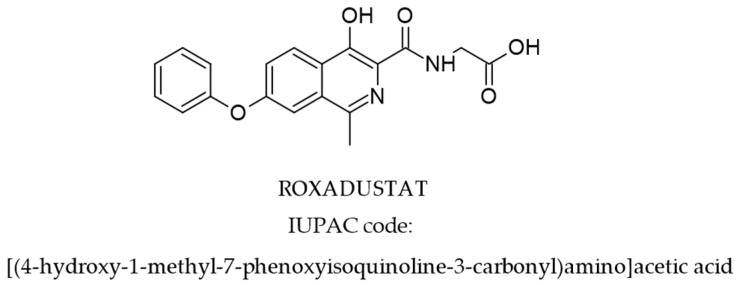
Structure of ROX [27].

**Figure 2 cimb-47-00734-f002:**
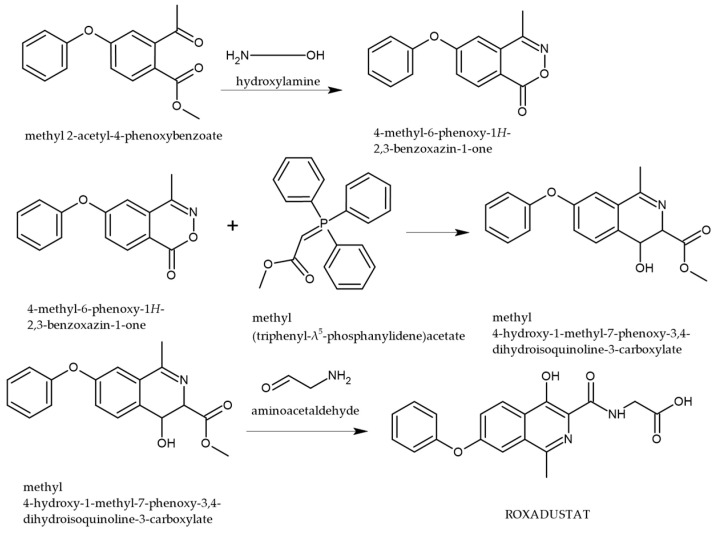
Synthetic pathway for the preparation of ROX [28].

**Figure 3 cimb-47-00734-f003:**
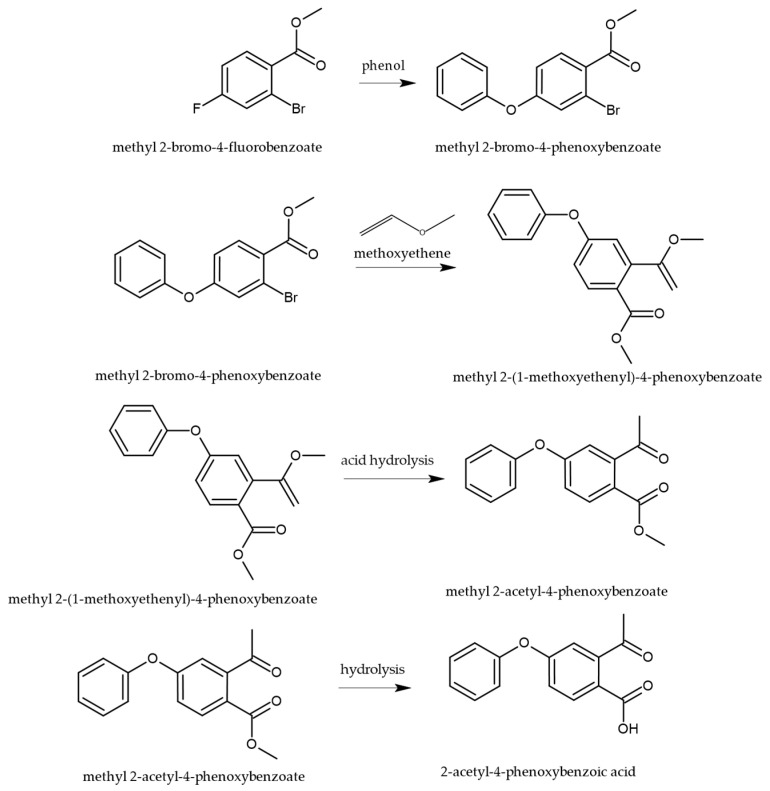
Synthetic pathway for the preparation of the key intermediate in the synthesis of ROX, 2-acetyl-4-phenoxy benzoic acid [28].

**Figure 4 cimb-47-00734-f004:**
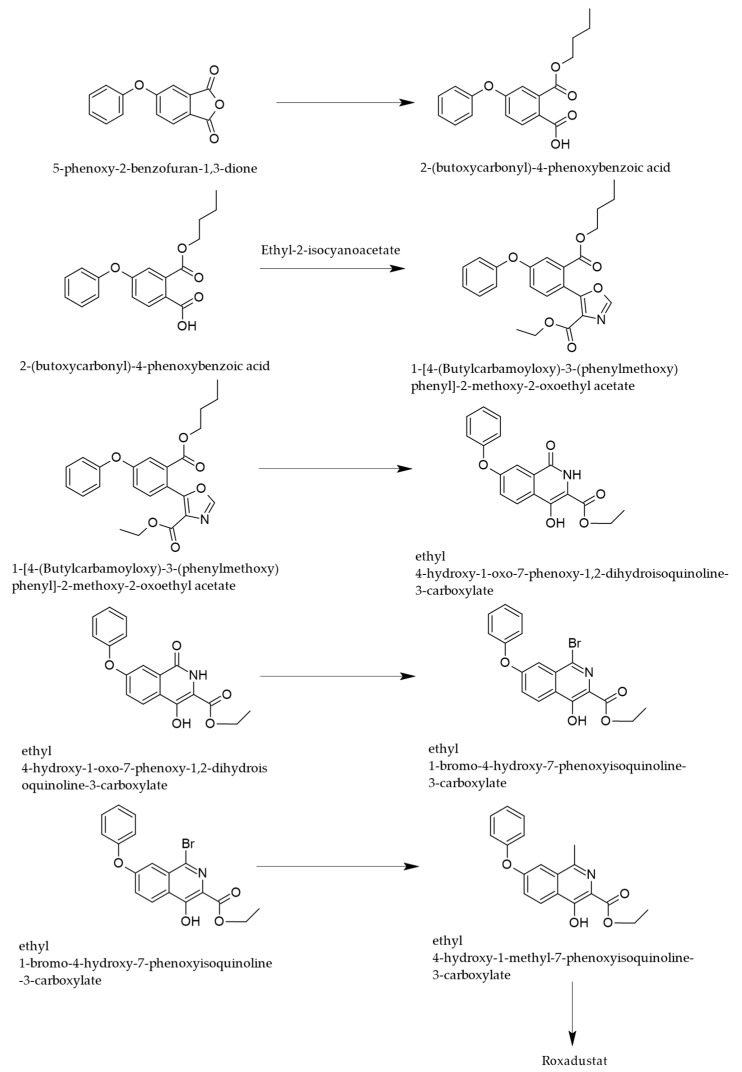
Synthetic pathway of ROX starting from phthalic anhydride derivatives [29].

**Figure 5 cimb-47-00734-f005:**
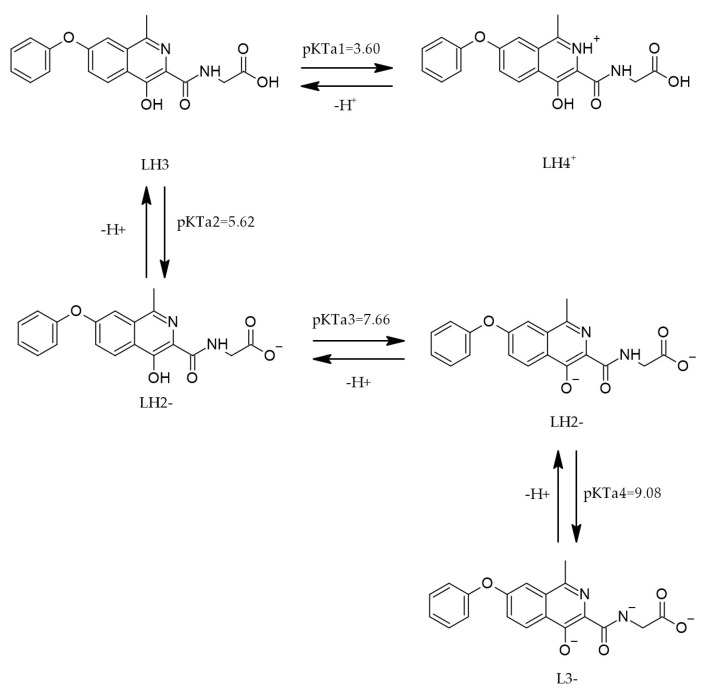
The protonation of ROX [30].

**Figure 6 cimb-47-00734-f006:**
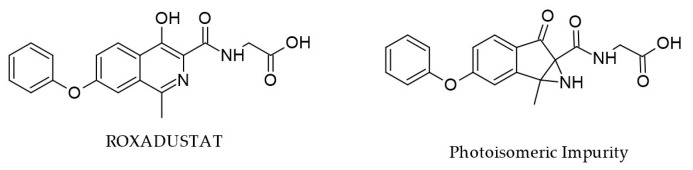
Chemical structures of ROX and PI [31].

**Figure 7 cimb-47-00734-f007:**
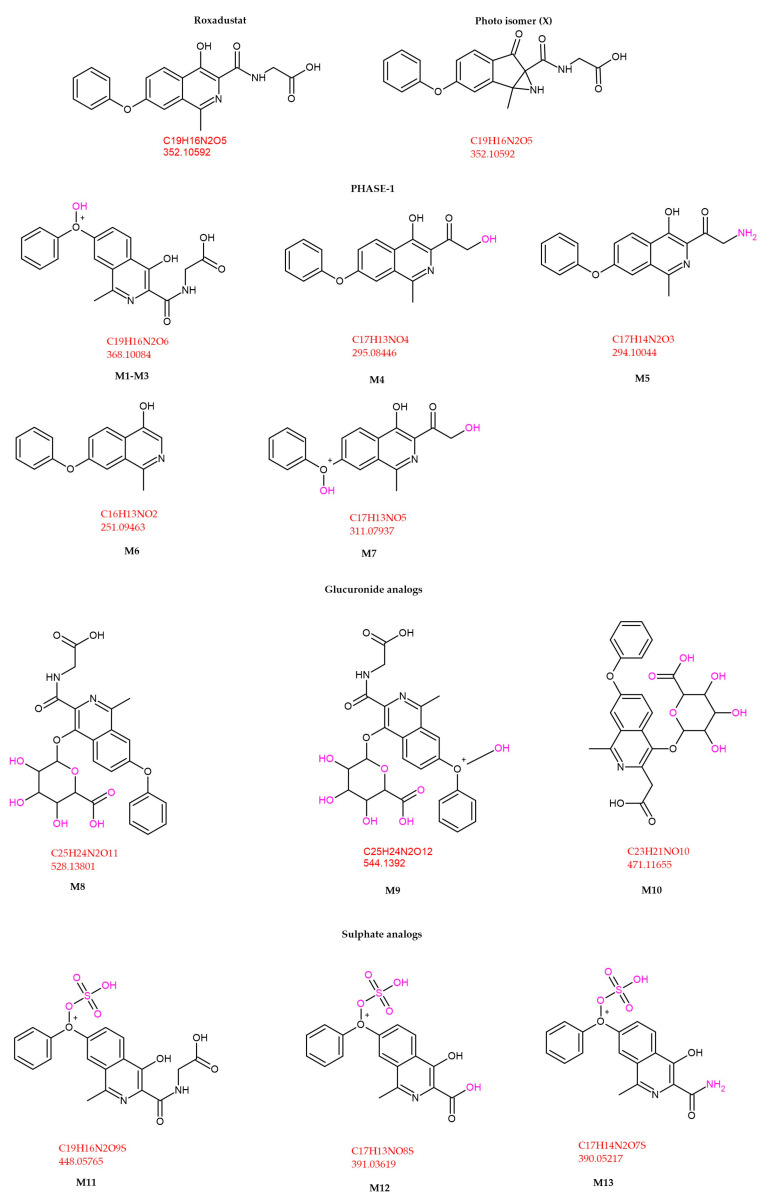
ROX metabolite structures detected in equine urine after a single oral dose [37].

**Figure 8 cimb-47-00734-f008:**
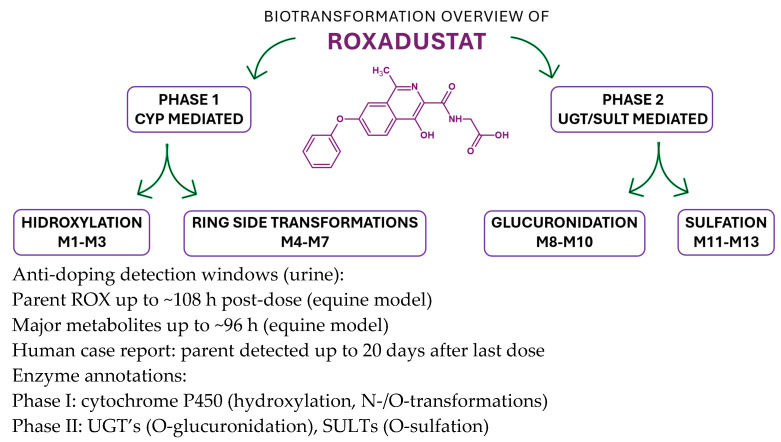
Biotransformation map of roxadustat (ROX) integrating Phase I and Phase II steps with enzyme-class annotations and anti-doping implications [38].

**Table 1 cimb-47-00734-t001:** Identification elements of ROX recorded by EMA [26].

Identification Element	Details
Registered Product Code (EMA)	EMEA/H/C/004871 [16]
Active pharmaceutical ingredient	Roxadustat
Global non-exclusive name	Roxadustat
Therapeutic uses	Anemia, nephrological chronic diseases
ATC Code (anatomical therapeutic chemical)—classification of the drug according to the system/organ targeted by the therapeutic intake and chemical and pharmacological performances [26]	B03XA05
Additional monitoring	Drug under monitoring process

**Table 2 cimb-47-00734-t002:** pH and pKa values of ROX in different pH ranges [30].

Parameter	Value at 25 °C	Value at 37 °C	Method
pKa1	4.33 (09)	4.25 (09)	Potentiometric
pKa2	6.57 (11)	6.49 (10)	Potentiometric
pKa3	8.88 (05)	8.80 (06)	Potentiometric
pKa4	9.03 (04)	9.00 (05)	Potentiometric
pKa1 (UV-metric)	3.60 (04)		Spectrophotometric (SQUAD84, REACTLAB)
pKa2 (UV-metric)	5.62 (14)		Spectrophotometric (SQUAD84, REACTLAB)
pKa3 (UV-metric)	7.66 (16)		Spectrophotometric (SQUAD84, REACTLAB)
pKa4 (UV-metric)	9.08 (02)		Spectrophotometric (SQUAD84, REACTLAB)
ΔH^0^ (enthalpy of dissociation)	Endothermic		
ΔG^0^ (Gibbs free energy)	Positive		
ΔS^0^ (entropy of dissociation)	Negative		
Identified protonated species	LH_4_^+^, LH_3_, LH_2_^−^, L_3_^−^	LH_4_^+^, LH_3_, LH_2_^−^, L_3_^−^	

**Table 3 cimb-47-00734-t003:** Equipment used for detection of ROX’s main metabolites [37].

Analysis	Equipment and Materials
LC	UltiMate 3000 UPLC+ (Dionex, Sunnyvale, CA, USA)
LC	Reversed-phase HPLC column: C18, 4.5 × 150 mm
LC	Eluents: A—5 mM ammonium acetate, 0.2% formic acid aqueous solution; B—cyanomethane
MS	Dionex UltiMate 3000 UHPLC+; QExactive high-resolution accurate mass spectrometer
MS	70,000 resolutions; mass range *m*/*z* 50–750.
MS	Capillary voltage—4000 V; capillary temperature—320 °C

**Table 4 cimb-47-00734-t004:** Retention times (RTs) of ROXs’ identified metabolites in equine urine following oral administration [37].

Analyte	Formula	RT [min]
ROX	C_19_H_17_N_2_O_5_	11.04
M1	C_19_H_17_N_2_O_6_	8.82
M2	C_19_H_16_N_2_O_6_	7.53
M3	C_19_H_16_N_2_O_6_	9.82
M4	C_17_H_13_NO_4_	8.03
M5	C_17_H_14_N_2_O_3_	11.05
M6	C_16_H_13_NO_2_	11.82
M7	C_17_H_13_NO_5_	10.99
M8	C_25_H_24_N_2_O_11_	7.56
M9	C_25_H_24_N_2_O_12_	7.15
M10	C_23_H_21_NO_10_	8.38
M11	C_19_H_16_N_2_O_9_S	7.74
M12	C_17_H_13_NO_8_S	7.33
M13	C_17_H_14_N_2_O_7_S	8.34

**Table 5 cimb-47-00734-t005:** LC-MS parameters of ROX [45].

Name	Roxadustat
Formula	C19H16N2O5
Polarity	+
*m*/*z*	353.113
RT (min.)	6.3
Manufacturer	MCE

**Table 6 cimb-47-00734-t006:** Major applications of roxadustat (ROX): evidence level, mechanisms, and risk–benefit overview.

Application/Context	Evidence Level	Key Mechanisms/Targets	Reported Benefits (Signal/Outcomes)	Noted Risks/Limitations	Risk–Benefit Summary	Evidence Gaps/Next Steps	Ref.
Calcium/Vitamin D axis (FGF23)	Clinical/observational in ESKD	↑ FGF23; SPP24 changes	Links to mineral metabolism markers	Possible calcification risk in ESKD	Uncertain; monitor minerals	Mechanistic and outcome studies	[67,68]
Iron deficiency/inflammation	Translational/ clinical mechanistic	HIF activation despite cytokines; ↓ hepcidin; iron mobilization	Maintains Hb increase in inflammatory milieu	Benefit depends on iron stores; risk of functional iron deficiency	Potentially favorable with iron monitoring	Prospective, trials stratified by CRP/hepcidin	[68,69,70,71,72]
PCOS (metabolic)	Preclinical (letrozole mice)	HIF-mediated metabolic re-programming	Improved metabolic profile (models)	No human validation	Unknown; exploratory	Translational metabolic studies	[72]
HF with anemia/inflammation	Hypothesis/limited clinical	HIF-mediated iron handling; anti-inflammatory effects	Potential improvement in iron indices/Hb	Cardiovascular safety needs careful monitoring	Equivocal; requires targeted trials	HF-specific RCTs with iron/hepcidin strata	[73,74]
CKD anemia (erythropoiesis)	Phase 3 RCTs /pooled analyses	HIF-PHD inhibition → HIF-1/2α stabilization; ↑ EPO; ↓ hepcidin; ↑ DMT1/ferroportin	↑ Hb; ESA noninferiority; reduced IV iron use in some settings	GI events, hypertension, hyperkalemia; thyroid function changes in dialysis subsets	Favorable in indicated CKD populations with monitoring	Head-to-head long-term CV outcomes by phenotype; thyroid signal characterization	[1,2,51,63,64,65,71,74,75,76,77]
Diabetic kidney disease (early)	Preclinical/cellular	HIF-1α–p53/p21 modulation; anti-proliferative in mesangial cells	Nephroprotective signals (preclinical)	Clinical translation unproven	Uncertain; hypothesis-generating	Early-phase trials with renal endpoints	[62,73,78]
Hemodialysis–thyroid axis	Observational/case reports	Central hypothyroidism; THR-β modulation	Reversible TSH/TRH changes reported	Potential endocrine signal	Caution; monitor thyroid	Prospective thyroid function monitoring	[79,80]
Chemotherapy Induced Anemia	Clinical-Phase 2 RCTs	HIF pathway activation	Increased Hb levels, reduced transfusion needs	Short-term study, with potential side effects and no control group	Increases Hb, reduces transfusions, evidence is limited by small design	Larger, long-term, randomized trials needed to confirm safety, efficacy and optimal dosing in diverse patients	[81]
Pure red cell aplasia (PRCA)	Case series/clinical use	Endogenous EPO stimulation	Restoration of Hb with immunosuppression	Heterogeneous etiologies; small N	Cautiously favorable in select cases	Registries/prospective cohorts	[82]
Intestinal inflammation	Preclinical, mouse models and B cells in vitro	HIF-1α regulates B cell metabolism and acetyl-CoA- dependent epigenetic changes to improve IgA class switching and to reduce intestinal inflammation	Enhanced IgA production, reduced severity of intestinal inflammation (DSS-induced colitis)	Human relevance is untested	Enhanced IgA production, reduced intestinal inflammation, effects in humans remain uncertain	Need for validation in human studies	[83]
Kidney stone disease	Preclinical (models of Ca-oxalate)	↓ CCL2, TNF, ADGRE1 (anti-inflammatory)	↓ Crystal deposition; renal protection in models	Human data lacking	Unknown; preclinical only	Pilot clinical feasibility	[84,85]
Water balance/edema	Preclinical	HIF-dependent; AQP1 regulation in proximal tubule	Mechanistic rationale for diuresis modulation	No clinical outcome data	Unknown; mechanistic only	Human biomarker studies	[86]
Immune thrombocytopenia (ITP)	Case/early clinical signals	HIF-1α immunomodulation; Hb elevation	Hb increase in ITP setting	Limited sample sizes; mechanism indirect	Unclear; adjunctive role at most	Controlled trials	[87,88,89]
Hemodialysis due to PRCA	Case series/clinical use	Endogenous EPO stimulation	PRCA treatment showed a positive response to rhEPO, cyclosporine and ROX	Single patient case report, diagnostic uncertainty	Effective PRCA treatment in one patient, potential therapy risks	Requires larger studies, long-term outcomes and improved diagnostic for PRCA	[82,90]
Peritoneal dialysis (PD) anemia	Phase 3 pooled analyses	EPO/HIF axis activation	Noninferior to ESA; safe in PD cohorts	Class-typical AEs; PD-specific data still limited	Favorable with standard monitoring	Longer follow-up in PD	[65,91]
Non-dialyzed patients with or without diabetes	Retrospective observational study	HIF pathway activation	Improved Hb, no difference in adverse effects between diabetic and non-diabetic patients	Small retrospective study, limited follow up	Effective in anemia treatment with dose-dependent risks	Larger, prospective, multicenter trials	[92]
Cardiorenal combo (with dapagliflozin)	Preclinical (CRS models)	PI3K/AKT/mTOR; ERK; JNK/p38 activation	Heart–kidney protection signals (models)	No human data	Unknown; preclinical synergy	Phase 1/2 PK-PD and safety	[93]
Tumor angiogenesis (TIMAP/SMAD)	Preclinical	↑ TIMAP via SMAD1/5/8 inhibition	Pro-angiogenic effects (context-specific)	Oncologic risk concern in tumors	Potential risk in active malignancy	Tumor-context contraindication guidance	[20,21,94]
Post-transplant anemia	Clinical study, level 2b	HIF pathway activation	Increased Hb, improved iron utilization, potential to reduce ESA use	Limited sample, population specific	Effective in anemia correction, improved iron metabolism, good short-term tolerability	Need larger, long term-studies, ESA comparison to confirm safety, efficacy and optimal dosing	[95]
GI GvHD/barrier protection	Preclinical (transplant models)	↓ IFN-γ, TNF-α; epithelial protection	Reduced early gut injury (models)	Clinical translation pending	Unknown; promising preclinical	Early clinical trials	[22,82,96,97]
Ischemic stroke/oxidative stress	Preclinical	HIF-1/NRF2 antioxidant activation	Neuroprotection; ↓ ROS in models	Timing/ dose critical; human data absent	Unknown; time-sensitive	Phase 1 neuroprotection studies	[49,93,98]
Polycystic Ovary Syndrome	Preclinical, mouse models of PCOS	HIF pathway activation	Improved glucose tolerance, improved insulin sensitivity, adipose tissue function	Preclinical setting, lack of human outcomes	Significant metabolic effects in PCOS mice	Human translation, long-term safety, reproductive outcomes remain the main evidence gaps	[99]
Diabetes/skeletal muscle	Preclinical (human myotubes)	↑ Glycolysis; ↓ mitochondrial respiration; ↑ glycogen synthesis	Improved insulin-stimulated glycogen synthesis (ex vivo)	In vivo human effect unknown	Unknown; mechanism-oriented	Physiology studies in T2D	[72,73,100]
Reversible central hypothyroidism	Clinical study, Level 3	Suppresses pituitary TSH production, leading to central hypothyroidism	Reversibility of thyroid dysfunction	Reversible central hypothyroidism, lack of clinical symptom data	Effective anemia treatment in hemodialysis patients, clinicians must monitor thyroid function due to the risk of potential central hypothyroidism	More robust, long-term studies	[101]
Anti-doping (detection)	Analytical validation/case report	Long-window urinary detection; metabolite panels	Detection at sub-pg/mL for extended periods; ABP integration	Not a therapeutic application	N/A (forensic context)	Harmonize target metabolites and cut-offs	[19,45,70,102]

Abbreviations: CKD, chronic kidney disease; PD, peritoneal dialysis; ESA, erythropoiesis-stimulating agent; HF, heart failure; GvHD, graft-versus-host disease; ITP, immune thrombocytopenia; PCOS, polycystic ovary syndrome. Note: Evidence level follows hierarchy: Phase 3 RCTs > pooled/observational > translational/mechanistic > preclinical. Risk–benefit summaries are qualitative and should be interpreted within labeled indications and patient-specific factors. Symbols: ↑: increased, elevated, upregulated; ↓: decreased, reduced, downregulated.

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
