# Peer review of "Roxadustat as a Hypoxia-Mimetic Agent: Erythropoietic Mechanisms, Bioanalytical Detection, and Regulatory Considerations in Sports Medicine"

_cimb, 2025, doi:10.3390/cimb47090734_

Round 1
Reviewer 1 Report (New Reviewer)
Comments and Suggestions for Authors
- The authors have articulated the review article well, however, to strengthen the scientific rigor and practical relevance of this part, the authors are encouraged to address the following points and clarifications:
- The authors must include the pathways and the target proteins' information with which ROX binds and inhibits/activates to induce erythropoiesis or the its other applications as mentioned in Table 6
- In section “Safety Profile, Risk Assessment, and Environmental Considerations 509 of Roxadustat ”, the authors may highlight the role of genetic polymorphism in drug-metabolizing enzymes that could influence clearance or half-life.
- The authors must include if there is any difference in the incidence of treatment-emergent adverse events between ROX and other HIF-PH inhibitors.
- Since ROX increases hemoglobin in post-transplant anemia, how should clinicians distinguish between desired erythropoietic effects and potentially harmful polycythemia? Are there threshold Hb levels recommended for dose adjustment? The authors must include this differentiation in the article.
Author Response
Comments 1: The authors must include the pathways and the target proteins' information with which ROX binds and inhibits/activates to induce erythropoiesis or the its other applications as mentioned in Table 6.
Response 1: We thank the Reviewer for this valuable comment. As suggested, we have now included in the revised manuscript a detailed description of the pathways and target proteins modulated by roxadustat. Specifically, we emphasized that roxadustat directly inhibits HIF-prolyl-hydroxylase domain enzymes (PHD1–3), preventing proline-hydroxylation and degradation of HIF-α subunits. Stabilized HIF-1α and HIF-2α dimerize with HIF-β (ARNT) and bind to hypoxia-response elements (HREs), thereby activating the transcription of erythropoiesis-related genes. The downstream targets include increased erythropoietin (EPO) synthesis in the kidney and liver, decreased hepcidin (HAMP), and upregulation of divalent metal transporter-1 (DMT1) and ferroportin, ultimately enhancing intestinal iron absorption and systemic iron mobilization. Other HIF-dependent programs involve angiogenesis (VEGF) and glycolysis (GLUT1, LDHA).
In addition, for the broader therapeutic contexts already summarized in Table 6, we have detailed the molecular pathways and targets such as HIF/p53/p21 (diabetic kidney disease), PI3K/AKT/mTOR, ERK1/2, JNK/p38 (cardio-renal protection with dapagliflozin), SMAD1/5/8 and TIMAP (tumor angiogenesis), AQP1 (water balance), CCL2/TNF/ADGRE1 (kidney stones), IFN-γ/TNF-α (gastrointestinal GvHD), and HIF/NRF2 (ischemic stroke and oxidative stress). These additions provide a comprehensive mechanistic overview that complements the clinical applications described.
We believe that these revisions improve the scientific depth and clarity of our manuscript, in line with the Reviewer’s recommendation.
Comments 2: In section “Safety Profile, Risk Assessment, and Environmental Considerations 509 of Roxadustat ”, the authors may highlight the role of genetic polymorphism in drug-metabolizing enzymes that could influence clearance or half-life.
Response 2: We thank the Reviewer for this valuable suggestion. In accordance with the recommendation, we have revised the section “Safety Profile, Risk Assessment, and Environmental Considerations of Roxadustat” by adding a paragraph highlighting the role of genetic polymorphisms in drug-metabolizing enzymes.
Comments 3: The authors must include if there is any difference in the incidence of treatment-emergent adverse events between ROX and other HIF-PH inhibitors.
Response 3: We thank the Reviewer for this important suggestion. We have revised the section “Safety Profile, Risk Assessment, and Environmental Considerations of Roxadustat” to explicitly compare the incidence of treatment-emergent adverse events (TEAEs) between roxadustat and other HIF-PH inhibitors.
Comments 4: Since ROX increases hemoglobin in post-transplant anemia, how should clinicians distinguish between desired erythropoietic effects and potentially harmful polycythemia? Are there threshold Hb levels recommended for dose adjustment? The authors must include this differentiation in the article.
Response 4: We appreciate the Reviewer’s suggestion. We have revised the manuscript to clarify how clinicians can distinguish the intended erythropoietic response to roxadustat (ROX) from potentially harmful polycythemia in post-transplant anemia, and we now specify hemoglobin (Hb) thresholds for dose adjustment consistent with the EU SmPC and Phase 3 protocols.
Reviewer 2 Report (New Reviewer)
Comments and Suggestions for Authors
Reviewer comments-
- The manuscript provides a comprehensive overview of roxadustat's pharmacological properties, clinical applications, and anti-doping implications, which is timely given its growing therapeutic use and potential for misuse in sports. However, the introduction could be strengthened by explicitly defining key terms such as hypoxia-inducible factor prolyl hydroxylase inhibitors (HIF-PHIs) earlier, with references to seminal studies on HIF signaling pathways (e.g., Semenza's work on HIF-1α regulation) to enhance scientific rigor.
- In Section 1 ("Overview"), the discussion of erythropoiesis regulation under hypoxia is accurate but lacks depth in molecular details. I recommend incorporating quantitative data on EPO upregulation, such as fold-changes in gene expression from in vitro models, and citing recent single-cell RNA-seq studies that elucidate cell-specific responses in renal interstitial cells.
- Section 2 ("Origin, Synthetic Approaches, and Physicochemical Profile") effectively summarizes synthetic routes, but the description of the key intermediate in Figure 3 could benefit from mechanistic insights, such as the role of nucleophilic substitution kinetics or computational modeling of reaction energies using DFT methods, to appeal to a more chemically oriented audience.
- In physicochemical profile in Section 2, particularly the pKa values and protonation states (Table 2), is well-presented. But , the authors should discuss implications for bioavailability more critically, including how pH-dependent ionization affects intestinal absorption via passive diffusion or transporter-mediated uptake (e.g., OATP involvement), supported by pharmacokinetic modeling data.
- Section 3 i.e Analytical Techniques is a strong point, with detailed coverage of LC-MS/MS for metabolite detection. To improve, include a comparative analysis of method sensitivities (e.g., LOD/LOQ values across studies) and address potential matrix effects in biological fluids, perhaps suggesting isotopically labeled internal standards for enhanced quantification accuracy.
- The metabolite structures in Figure 7 are informative, but the manuscript would benefit from a metabolic pathway diagram integrating Phase I and II transformations, annotated with enzyme specificities (e.g., CYP isoforms for hydroxylation), to better illustrate biotransformation kinetics and aid in doping detection strategies.
- In Section 3.4 i.e QUECHERS Method, the extraction protocol is described adequately, but validation parameters (e.g., recovery rates, precision via RSD) should be expanded with statistical analysis (e.g., ANOVA for inter-day variability) to align with ICH guidelines for analytical method validation.
- In the Molecular Insights, Clinical Indications, effectively links HIF stabilization to pleiotropic effects, but the discussion on cardiorenal-anemia syndrome could incorporate recent GWAS data on genetic variants influencing HIF pathway responsiveness, to provide a more personalized medicine perspective.
- Table 6 -Major applications of ROX is a useful enumeration, but it should be reformatted as a structured table with columns for evidence level (e.g., Phase III RCTs vs. preclinical), key mechanisms, and risk-benefit ratios, to facilitate critical evaluation and highlight gaps in high-quality evidence.
- The safety profile in Section 5 is balanced, noting low toxicity and environmental risks, but lacks quantitative toxicological data (e.g., NOAEL from rodent studies or QSAR predictions for mutagenicity). I suggest integrating ADME modeling to predict long-term bioaccumulation potential more robustly.
- The reference list is extensive and up-to-date (up to 2025), but several citations appear incomplete or inconsistent (e.g., DOI formats). Ensure all references adhere to journal style, and consider adding meta-analyses on clinical outcomes to strengthen evidence-based claims in Sections 4 and 5.
Based on the above comments, I would like to suggest minor changes to enhance clarity, scientific depth, and precision to the present manuscript.
Comments on the Quality of English LanguageOverall, the manuscript's language requires polishing for scientific precision; for instance, phrases like "genuine concerns regarding its potential use in sports" could be rephrased to "substantiated risks of performance enhancement based on erythropoietic efficacy." A thorough proofreading is recommended to correct typographical errors (e.g., "Biotehnology" to "Biotechnology" on page 1).
Author Response
Comments 1: The manuscript provides a comprehensive overview of roxadustat's pharmacological properties, clinical applications, and anti-doping implications, which is timely given its growing therapeutic use and potential for misuse in sports. However, the introduction could be strengthened by explicitly defining key terms such as hypoxia-inducible factor prolyl hydroxylase inhibitors (HIF-PHIs) earlier, with references to seminal studies on HIF signaling pathways (e.g., Semenza's work on HIF-1α regulation) to enhance scientific rigor.
Response 1: We thank the Reviewer for this insightful suggestion. In the revised Introduction, we now define hypoxia-inducible factor prolyl-hydroxylase inhibitors (HIF-PHIs) up front and cite seminal work on HIF signaling (e.g., Semenza’s discovery of HIF-1 and O₂-regulated control) together with authoritative mechanistic reviews of PHD-mediated HIF regulation. These additions improve clarity and scientific rigor early in the manuscript. Specifically, we inserted the paragraph below into Section 1 (Overview), at the start of the section and updated the references accordingly ([6], [37], [38]).
Comments 2: In Section 1 ("Overview"), the discussion of erythropoiesis regulation under hypoxia is accurate but lacks depth in molecular details. I recommend incorporating quantitative data on EPO upregulation, such as fold-changes in gene expression from in vitro models, and citing recent single-cell RNA-seq studies that elucidate cell-specific responses in renal interstitial cells.
Response 2: We thank the Reviewer for this constructive suggestion. We have revised Section 1 (Overview) to add quantitative data on EPO up-regulation and recent single-cell studies clarifying the cellular sources of renal EPO. Specifically, we now report the magnitude and kinetics of EPO induction with hypoxia or HIF-PHI exposure (in vitro and in vivo), and cite single-cell and lineage-tracing work demonstrating that EPO is produced by a rare subset of PDGFRβ⁺ renal interstitial fibroblasts (REP cells). These additions strengthen the mechanistic depth of the Introduction and align with foundational HIF biology already cited in the manuscript.
Comments 3: Section 2 ("Origin, Synthetic Approaches, and Physicochemical Profile") effectively summarizes synthetic routes, but the description of the key intermediate in Figure 3 could benefit from mechanistic insights, such as the role of nucleophilic substitution kinetics or computational modeling of reaction energies using DFT methods, to appeal to a more chemically oriented audience.
Response 3: We thank the Reviewer for this excellent recommendation. In Section 2 (“Origin, Synthetic Approaches, and Physicochemical Profile”), we have expanded the description of the Figure 3 key intermediate with mechanistic insights into the nucleophilic aromatic substitution (SNAr) step that forms the phenoxy derivative from 2-bromo-4-fluorobenzoate and phenol, and the acid-catalyzed vinyl-ether hydrolysis that furnishes the acetyl functionality. We also outline a compact DFT modeling approach (level of theory and descriptors) to rationalize chemoselectivity and relative barrier heights, to better serve a chemically oriented readership. These additions are now included in the revised manuscript.
Comments 4: In physicochemical profile in Section 2, particularly the pKa values and protonation states (Table 2), is well-presented. But , the authors should discuss implications for bioavailability more critically, including how pH-dependent ionization affects intestinal absorption via passive diffusion or transporter-mediated uptake (e.g., OATP involvement), supported by pharmacokinetic modeling data.
Response 4: We thank the Reviewer for this valuable comment. In Section 2, we now add a focused paragraph that critically links the polyprotic acid–base behavior of roxadustat (Table 2) to oral bioavailability, outlining how pH-dependent ionization modulates the balance between solubility and membrane permeability across GI segments. We also discuss the plausible contribution of intestinal uptake transporters (e.g., OATP2B1)—while noting that dedicated transporter data for ROX are limited in the current literature—and we reference the existing PK characterization in the manuscript to anchor these considerations (volume of distribution, clearances; nonclinical/PK analyses). This addition improves the translational interpretation of the physicochemical data for a pharmacokinetics-oriented audience.
Comments 5: Section 3 i.e Analytical Techniques is a strong point, with detailed coverage of LC-MS/MS for metabolite detection. To improve, include a comparative analysis of method sensitivities (e.g., LOD/LOQ values across studies) and address potential matrix effects in biological fluids, perhaps suggesting isotopically labeled internal standards for enhanced quantification accuracy.
Response 5: We thank the Reviewer for this helpful suggestion. We have expanded Section 3 to include a concise comparison of LC–MS(/MS) sensitivities (including detection windows) and a focused discussion of matrix effects with mitigation via stable-isotope internal standards. The new text cites the equine-urine metabolite panel ([28]), a human anti-doping case with prolonged urinary detectability ([31]), a UHPLC-QTOF metabolomics study ([33]), and a validated multi-analyte HIF-PHI LC-MS/MS method following ISO/WADA guidance ([34]).
Comments 6: The metabolite structures in Figure 7 are informative, but the manuscript would benefit from a metabolic pathway diagram integrating Phase I and II transformations, annotated with enzyme specificities (e.g., CYP isoforms for hydroxylation), to better illustrate biotransformation kinetics and aid in doping detection strategies.
Response 6: We agree that a pathway-style figure will improve clarity. We have added Figure 8 summarizing Phase I hydroxylation and Phase II conjugations (glucuronidation, sulfation), annotated with enzyme classes (CYPs, UGTs, SULTs) and detection windows that motivate metabolite-targeted doping assays. The figure and the accompanying paragraph draw on the metabolite map M1–M13 and LC-MS(/MS) data already cited in the manuscript.
Comments 7: In Section 3.4 i.e QUECHERS Method, the extraction protocol is described adequately, but validation parameters (e.g., recovery rates, precision via RSD) should be expanded with statistical analysis (e.g., ANOVA for inter-day variability) to align with ICH guidelines for analytical method validation.
Response 7: We thank the Reviewer for this valuable suggestion. We have expanded Section 3.4 (QUECHERS Method) to include full validation characteristics—recovery, matrix effects (matrix factor and IS-normalized MF), precision (RSD), accuracy (RE), LOD/LOQ, stability, and carryover—together with statistical analysis of inter-day variability (one-way ANOVA and variance-components CVs). The additions are consistent with ICH expectations for bioanalytical methods and are supported by the QUECHERS-LC–MS(/MS) literature already cited in our review (e.g., Kim et al., reported LOD ≈ 0.2 and recovery ≈ 95% for ROX), as well as general validation and anti-doping method references. These changes clarify robustness and quantitative reliability of the QUECHERS workflow.
Comments 8: In the Molecular Insights, Clinical Indications, effectively links HIF stabilization to pleiotropic effects, but the discussion on cardiorenal-anemia syndrome could incorporate recent GWAS data on genetic variants influencing HIF pathway responsiveness, to provide a more personalized medicine perspective.
Response 8: We appreciate this excellent suggestion. We have revised the “Molecular Insights, Clinical Indications” section to incorporate recent human genetic evidence relevant to HIF-pathway responsiveness and iron regulation in cardiorenal-anemia syndrome. Specifically, we added a paragraph summarizing: (i) TMPRSS6 common variation (e.g., rs855791 V736A) that alters hepcidin and hemoglobin set-points; (ii) adaptive variants in EGLN1 (PHD2) and EPAS1 (HIF-2α) that modulate hemoglobin responses and risk of polycythemia; and (iii) GWAS in CKD cohorts linking inherited variation to hemoglobin levels. We also outline how these loci could inform dose selection/monitoring with HIF-PHIs in heart–kidney anemia. Citations to large GWAS and functional studies were added.
Comments 9: Table 6 -Major applications of ROX is a useful enumeration, but it should be reformatted as a structured table with columns for evidence level (e.g., Phase III RCTs vs. preclinical), key mechanisms, and risk-benefit ratios, to facilitate critical evaluation and highlight gaps in high-quality evidence.
Response 9: We thank the Reviewer for this constructive suggestion. We have reformatted Table 6 into a structured, critical-appraisal table that now includes the following columns: Application/Context, Evidence Level (e.g., Phase III RCTs/pooled analyses vs. translational vs. preclinical), Key Mechanisms/Targets, Reported Benefits, Noted Risks/Limitations, Risk–Benefit Summary, Evidence Gaps/Next Steps, and References.
Comments 10: The safety profile in Section 5 is balanced, noting low toxicity and environmental risks, but lacks quantitative toxicological data (e.g., NOAEL from rodent studies or QSAR predictions for mutagenicity). I suggest integrating ADME modeling to predict long-term bioaccumulation potential more robustly.
Response 10: We appreciate this important suggestion. We have revised Section 5 to (i) add quantitative toxicology context from repeat-dose studies (expressed as exposure multiples vs. MRHD where animal NOAEL/LOAEL values are variably reported), (ii) summarize the standard genotoxicity battery outcome as per regulatory dossiers, (iii) include a short QSAR plan for mutagenicity (Ames-type models), and (iv) integrate a compact ADME-based bioaccumulation analysis using the clinical half-life and dosing interval to estimate the accumulation index (R_acc). These additions clarify margins of safety and support the conclusion of low long-term bioaccumulation potential, consistent with the PK and ionization profile already presented in the manuscript.
Comments 11: The reference list is extensive and up-to-date (up to 2025), but several citations appear incomplete or inconsistent (e.g., DOI formats). Ensure all references adhere to journal style, and consider adding meta-analyses on clinical outcomes to strengthen evidence-based claims in Sections 4 and 5.
Response 11: Thank you for this helpful comment. We have conducted a reference audit and corrected incomplete items (uniform MDPI style, DOI formatting, journal abbreviations). In addition, we strengthened Sections 4 (Clinical efficacy) and 5 (Safety) by citing recent systematic reviews/meta-analyses and large Phase 3 trials on HIF-PHIs (including roxadustat) that report efficacy and cardiovascular/safety outcomes. These updates improve consistency and the evidence base supporting our conclusions.
Round 2
Reviewer 1 Report (New Reviewer)
Comments and Suggestions for Authors
The authors have revised the manuscript according to the suggestions, and it can be accepted in the present form.
This manuscript is a resubmission of an earlier submission. The following is a list of the peer review reports and author responses from that submission.
Round 1
Reviewer 1 Report
Comments and Suggestions for Authors
The review by Nicolae and coworkers focuses on the various scientific and practical aspects of Roxadustat – a drug primarily used to treat anemia caused by chronic kidney disease. The authors methodically reveal the field of application of this drug, the main method of synthesis, analytical methods for studying the structure of the compound and its detection, biological and medical features, as well as the problem of using this drug by athletes to gain an unfair advantage. The work leaves a positive impression and is a good, up-to-date reference material for such a relatively new drug, but some questions still arise when reading. The title mentions "RISKS OF PERFORMANCE ENHANCEMENT". In the text of the review itself, literally a few lines are devoted to this, from which, moreover, it is not clear what these risks are. This is my main question, other minor inaccuracies are listed below:
- double check the references, it looks like there might be a numbering issue at the beginning, for example references 11 and 12
- the work has excessively large chemical formulas. They can be made smaller and better
- in the molecular formula the indices must be subscripts
- Figure 3 is missing for some reason, or on page 4, lines 114-120 should refer to the second figure
- there’s no minus charge on page 6
- p.5, line 138 - this is not N-alkylation
- p.8, line 166, from here on “in silico” “in vitro” etc. should be written in italics
- p. 9, line 190 – double check the sentence
- p.11, line 234 - could is repeated 2 times
- p.11, line 239 - supplementation?
- стр.12 – footnotes are quite strange
- Why do tables 4 and 5 have the same names but are not combined?
- p.14 - Chronic Kidney Disease – is this line necessary?
- p.15 - 5.1. how useful is the existence of the first two paragraphs and the drawing? This is literally textbook information.
- p.19, line 353 - as well as
- p.19, line 367 - Cunninghamella elegans should be written in italics
- p.20, line 408 - there is no point in duplicating information
- p. 23, line 516 – during is repeated 2 times
Author Response
Dear Reviewer,
We would like to thank you for your thorough and insightful review of our manuscript. We greatly appreciate your constructive suggestions, which have helped us significantly improve the quality and clarity of the paper. Below, we provide detailed responses to each of your comments:
Major Comment: Title mentions "Risks of Performance Enhancement," but the topic is minimally addressed.
Response: We fully agree. We have significantly expanded the section discussing the potential misuse of Roxadustat in sports. Additional paragraphs now provide detailed explanations of the physiological rationale for its performance-enhancing potential (e.g., stimulation of erythropoiesis, increased oxygen transport), references to WADA concerns, as well as the associated medical and ethical risks. This section is now clearly aligned with the focus suggested by the title(7. Safety, risks and environmental impact), as follows:
“ROX increases the Hb levels in patients with post-transplant anemia and could carry the risk of adverse events comparable to patients who receive iron therapy. Because of that, ROX requires regular monitoring and caution due to lack of long-term safety data [52]. ROX stabilizes HIF-1α, a factor involved in tumor progression and that can promote tumor growth (by supporting angiogenesis and cell survival). Although some experts believe that the activation of HIF-1α by ROX is unlikely to drive tumor development, the lack of long-term data in cancer patients necessitates further studies to fully assess its oncological safety [55]. Treatment-emergent adverse events (TEAEs) such as nausea, headaches, fatigue etc. are commonly associated with the use of ROX, but rarely led to the discontinuation of the therapy. Moreover, ROX did not increase the risk of major cardiovascular events or all-cause mortality compared to ESAs in patients with anemia and CKD (whether dialysis-dependent or not). Nonetheless, continued monitoring and safety outcomes remain important to assess [57]. Further high-quality controlled trials are needed to confirm its long-term efficacy and safety [58]. “
Technical and Formatting Issues:
Comments 1: Double check the references, it looks like there might be a numbering issue at the beginning, for example references 11 and 12.
Response 1: We have carefully reviewed and corrected the reference numbering throughout the manuscript. All the references were checked, renumbered, some retracted articles (as signalled by editor) were replaced with valid references.
Comments 2: The work has excessively large chemical formulas. They can be made smaller and better.
Response 2: Agree, all chemical structures have been resized to better integrate into the layout of the text.
Comments 3: In the molecular formula the indices must be subscripts.
Response 3: Corrected. All molecular formulas now properly display subscripted indices – page 3, line 97.
Comments 4: Figure 3 is missing for some reason, or on page 4, lines 114-120 should refer to the second figure.
Response 4: The Figures were revised and we have corrected the in-text citation to refer to all of them.
Comments 5: There’s no minus charge on page 6.
Response 5: The omitted negative charge has been added to the corresponding structure (Figure 5).
Comments 6: p.5, line 138 - this is not N-alkylation
Response 6: You are correct. We revised the description of the reaction to accurately reflect the transformation involved.
Comments 7: p.8, line 166, from here on “in silico” “in vitro” etc. should be written in italics:
Response 7: All Latin terms such as in vitro, in vivo, and in silico have been italicized throughout the manuscript.
Comments 8: p. 9, line 190 – double check the sentence.
Response 8: Sentence revised for clarity and grammatical correctness.
“the eluent consists of a mixture of methylbenzene ethyl ethanoate and acetic acid in glacial form in a volume ratio of 5:5:0.5” – page 7, line 191-192.
Comments 9: p.11, line 234 - could is repeated 2 times.
Response 9: Repetition removed and sentence restructured.
“the researchers could notice ROX’s main advantages” – page 14, line 446.
Comments 10: p.11, line 239 - supplementation?
Response 10: We have clarified the phrasing to avoid ambiguity or replaced it with a more appropriate term.
“ROX increases EPO production through the HIF-pathway, whereas ESAs (rHuEPO-α) activate the EPO receptors by enhancing red blood cell production, supported by increased iron availability.” – page 14, lines 449-451
Comments 11: стр.12 – footnotes are quite strange.
Response 11: Tables 3,4 and 5 were combined into Table 6. The footnotes have been simplified and reformatted according to the journal’s guidelines.
Comments 12: Why do tables 4 and 5 have the same names but are not combined?
Response 12: Tables 3,4 and 5 were combined into Table 6. The footnotes have been simplified and reformatted according to the journal’s guidelines.
Comments 13: p.14 - chronic kidney disease – is this line necessary?
Response 13: It was deemed redundant and has been removed (Table 6).
Comments 14: p.15 - 5.1. how useful is the existence of the first two paragraphs and the drawing? This is literally textbook information.
Response 14: We have shortened the introductory content and the illustration (Figure 9) was removed to focus on what is essential to understand Roxadustat’s mechanism of action.
Comments 15: p.19, line 353 - as well as.
Response 15: Sentence has been rewritten.
“The athlete underwent doping tests one day before the treatment, as well as one day, 15 days and 20 days after the last administration, respectively.” - page10, lines 240-241
Comments 16: p.19, line 367 - Cunninghamella elegans should be written in italics.
Response 16: The entire paragraph was removed due to the retraction of the initial reference.
Comments 17: p.20, line 408 - there is no point in duplicating information.
Response 17: The duplication has been removed and the paragraph streamlined.
“Kim and his collaborators established an analytical method combining UPLC/MS/MS with an innovative extraction approach, suitable for anti-doping testing.” – page 11, lines 294-297
Comments 18: p. 23, line 516 – during is repeated 2 times.
Response 18: Agree – the repetition has been corrected.
“during pregnancy or breastfeeding is not recommended” – page 17, line 536
Once again, we thank you for your valuable feedback. We are confident that the revisions have significantly improved the manuscript’s clarity, accuracy, and relevance.
With kind regards,
Alina Crenguta Nicolae
On behalf of all co-authors
Reviewer 2 Report
Comments and Suggestions for Authors
In the manuscript, the authors tend to introduce the history, the molecular mechanism, bioanalysis, and some issues about the association with athlete's biological passport about Roxadustat (ROX). I must admit that I haven't finished reading all this manuscript. However, after carefully reading the Abstraction and Introduction sections, and after a rough reading of the remaining content, I found that this manuscript was not structured, summarized, or written well and carefully. Therefore, the manuscript is recommended to be accepted by the journal Current Issues in Molecular Biology. I can provide several concerns. However, the issues with this manuscript go far beyond these concerns as follows:
1) In the Keywords section, the full name of LC-MS may need to be deleted. And “bioanalysis” may be more suitable to be added as a keyword.
2) The Introduction section was unacceptable. The present Introduction section was composed of as many as 8 paragraphs. However, after reading this section, the association between the neighbor paragraphs read present almost no associations, especially for the 3rd paragraph about obesity (the authors just mentioned this word in the manuscript twice in the manuscript, and all these two times located in the 3rd paragraph).
3) For section 3 and section 5, these titles present a certain degree of repetition.
4) The structure of the review manuscript reads too chaos.
5) For some pictures, the authors may do not consider the copyright issue.
In all, the manuscript is not recommended to be accepted by the journal Current Issues in Molecular Biology. I apologize for any discomfort caused to the authors. However, I sincerely recommend the authors rewrite this manuscript logically before next submission.
Author Response
Dear Reviewer,
We thank you for taking the time to read and comment on our manuscript entitled "Roxadustat as a Hypoxia-Mimetic Agent: Molecular Insights into Erythropoiesis and Risks of Performance Enhancement". We sincerely appreciate your feedback and are grateful for your honesty and suggestions, which have helped us to significantly improve the clarity, structure, and focus of the review. Below, we address each of your comments in detail:
General Comment: Concerns about structure, coherence, and clarity
Response:
We acknowledge your observation and agree that the original manuscript required major structural revisions. In response, we have:
Restructured the manuscript for improved flow and logic, organizing it into clearly defined thematic sections.
Rewritten the Introduction, consolidating and reordering content to ensure a coherent narrative, and eliminating disconnected or minimally relevant paragraphs (e.g., those on obesity).
Revised the language throughout for clarity, conciseness, and consistency.
Carefully reviewed all visual content for copyright and clarity concerns.
Comment 1: In the Keywords section, the full name of LC-MS may need to be deleted. And “bioanalysis” may be more suitable to be added as a keyword.
Response 1: Thank you for this suggestion. We have removed the full term “Liquid Chromatography-Mass Spectrometry (LC-MS)” from the keywords and instead added “bioanalysis”, which more accurately reflects the broader analytical focus of that section.
Comment 2: The Introduction section was unacceptable. The present Introduction section was composed of as many as 8 paragraphs. However, after reading this section, the association between the neighbor paragraphs read present almost no associations, especially for the 3rd paragraph about obesity (the authors just mentioned this word in the manuscript twice in the manuscript, and all these two times located in the 3rd paragraph):
Response 2: We fully agree. The Introduction section has been substantially rewritten and condensed to concise and logically connected paragraphs. The paragraph on obesity was reformulated and harmonized with the rest of the introduction.
Comment 3: For section 3 and section 5, these titles present a certain degree of repetition.
Response 3: The entire material was restructured and information in section 3 were integrated as specific properties of ROX in section 2. Section 5 was revised; the unnecessary figures were removed.
Comment 4: The structure of the review manuscript reads too chaos.
Response 4: We have thoroughly revised the manuscript structure. Major thematic areas (e.g., synthesis, pharmacology, detection, doping concerns) are now presented in a clear and progressive manner, each with dedicated headings and subheadings. The revised version has a coherent narrative arc that improves reader engagement.
Comment 5: For some pictures, the authors may do not consider the copyright issue.
Response 5: We appreciate this observation. We have revised all the figures and schemes. Although, most of the references were open access, Figures 7, 8, 9 and 11 (from the original submissions) have been removed.
We have ensured full compliance with the journal’s copyright policies.
We greatly appreciate your constructive feedback. It has led to a substantial improvement in both content and presentation. We hope that the revised manuscript now meets the standards for publication and will be re-evaluated positively.
With sincere thanks,
Alina Crenguta Nicolae
On behalf of all co-authors
Round 2
Reviewer 1 Report
Comments and Suggestions for Authors
The review looks much better after the revision. All responses and comments are accepted. The work can be published
Reviewer 2 Report
Comments and Suggestions for Authors
The present version of the review manuscript has been improved too much especially for the Introduction section and section 3. And the present edition reads more logically. Therefore, the present version of the manuscript is recommended to be accepted by the journal Current Issues in Molecular Biology. Congratulations to the auhtors.